# Thermal Preload for Predicting Performance Change Due to Pad Thermal Deformation of Tilting Pad Journal Bearing

**Yon-Do Chun** [1], **Jiheon Lee** [1], **Jiyoung Lee** [1] and **Junho Suh** [2,*]

1   Electric Machines and Drives System Research Center, Korea Electrotechnology Research Institute, Changwon-si 51543, Republic of Korea
2   School of Mechanical Engineering, Pusan National University, Busan 46241, Republic of Korea
*   Correspondence: junhosuh@pusan.ac.kr; Tel.: +82-51-510-2332

**Abstract:** Thermal deformation of journal bearings operating under high-temperature conditions can have a significant effect on changes in bearing performance. However, no attempt has been made to quantify this amount of thermal deformation and link it to the performance change of the bearing. The aim of this study is to investigate the quantitative performance change due to thermal deformation of the tilting pad journal bearing (TPJB) pad in terms of the change in preload amount. The variable viscosity Reynolds equation and the energy equation were coupled using the relationship between viscosity and temperature, and the solution was obtained using the finite element method. Heat transfer between the spinning journal, oil film and pads is considered, and a three-dimensional (3D) finite element (FE) model was used to calculate the thermal deformation of the bearing structure. The steady state of the rotor-bearing system was predicted using a bearing performance prediction algorithm with three closed loops. State variables for this steady-state prediction include the amount of thermal deformation of the structure. In order to investigate the amount of thermal deformation of the bearing pad in terms of bearing performance, the concepts of thermal offset preload and thermal performance preload were suggested and the change in thermal preload under various conditions was investigated.

**Keywords:** tilting pad journal bearings; thermal preload; thermal deformation; dynamic performance; static performance

## 1. Introduction

Tilting pad journal bearings theoretically have cross coupled stiffness and damping terms close to zero, and due to these advantages, they are widely used in high-speed rotating machinery despite high manufacturing cost and machining difficulties. In addition, recent studies have been actively conducted to accurately predict the characteristics of bearings supporting the rotor due to turbomachinery, which is becoming lighter, faster, and higher in temperature [1,2].

Even if the bearing is manufactured as intended by the rotor-bearing designer, the bearing pad exhibits both elastic and thermal deformation during operation. However, it is impossible to measure both types of deformation under operating conditions. The elastic deformation is caused by the oil film pressure and pivot load applied to the bearing pad, whereas the thermal deformation is caused by the temperature distribution in the pad. Most of the heat source causing the thermal deformation of the pad is viscous shear in the oil film [3,4].

The amount of preload of journal bearings, along with the length-diameter (LD) ratio, is one of the most important design parameters for tilting pad journal bearings that a bearing designer should consider in the design stage [1,2]. Unlike that of rolling bearings, the preload of journal bearings is defined as a geometric parameter of the bearing pad. When the shape of the inner surface of the bearing pad is a circle with the same center of

curvature as the center point of the journal with zero eccentricity, the preload becomes zero. As the amount of preload increases, the radius of curvature of the inner surface of the pad increases. In general, the preload of journal bearings is zero or more. The thermal effect generated by journal bearings has been studied by many researchers, and the following are representative studies [3–43].

Reynolds equation for calculating the pressure of the oil film assumes that the viscosity is constant in the thickness direction of the oil film. Dowson [5] derived a generalized Reynolds equation considering the general condition in which the viscosity is not constant in the oil film thickness direction. Dowson et al. [6] constructed a journal bearing experimental device and studied the temperature characteristics of the bearing system under a static load. Tieu [7] discretized the three-dimensional energy equation using the finite element method. The temperature between the two plates was predicted under various conditions. In addition, several researchers [8–18] have studied the thermal properties of journal bearings through numerical methods and experiments.

Taniguchi et al. [19] presented a three-dimensional thermo-hydrodynamic (THD) lubrication model of a 19-inch diameter tilting pad journal bearing for a steam turbine with consideration of the laminar and turbulent flow regimes. The total heat balance based mixing inlet temperature was calculated to predict the pad inlet temperature. The predicted data including eccentricity, metal surface temperature and frictional loss were compared with experimental data.

Fillon et al. [20] conducted experiments and numerical analysis simultaneously on four pads tilting pad journal bearing with load-between-pads (LBP) characteristics. The bearing temperature was measured using 40 temperature sensors and compared with the numerical results.

Simmons and Dixon [21] performed an experimental study with 200 mm and five pads tilting pad journal bearing. The bearing angular position could be moved so that the load direction could be changed. They found that the load direction has a critical influence on the maximum pad temperature.

Kim et al. [22] presented a finite element approach-based two-dimensional tilting pad journal bearing model with a thermo-elasto-hydrodynamic (TEHD) lubrication model. A generalized Reynolds equation considering varying viscosity in fluid film thickness direction and an energy equation were coupled via a temperature dependent viscosity relation. The Reynolds boundary condition using a modified back-substitution procedure was adopted to take into account the zero pressure cavitation area. To avoid numerical oscillation problem occurring in energy equation, an up-winding scheme was employed. The classic mixing temperature theory was used to predict the fluid temperature flowing into the pad. Both elastic and thermal deformation pad were modeled. Hertzian contact theory-based pivot elastic deformation was modeled and included in the numerical bearing model. To obtain the static equilibrium state of the bearing system, the classical Newton–Raphson method was applied where equilibrium states include the film temperature distribution, pad temperature distribution, pad deformation, shaft temperature, sump temperature, pad tilt angles and journal position.

Kim et al. [23] presented a TEHD lubrication model-based tilting pad journal bearing numerical model. They adopted modal coordinate transformation to reduce the computation time for the analysis of the pad's elastic deformation. This was the first research where the modal displacement and velocity components were added into the bearing stiffness and damping coefficients. The simulation results were compared with earlier experimental works.

In the same year, Gadangi and Palazzolo [24] performed a time transient analysis of a tilting pad journal bearing system considering thermal effects and pad deformations. For the evaluation of the fluid film thickness, pad radial deformation was included in the film thickness equation. Three different lubricant models were studied; (a) isoviscous rigid, (b) isoviscous flexible pad, and (c) isoadi rigid pad cases. The thermal effects were found to have little effect on the bearing-journal dynamic behavior, whereas the pad deformation

had noticeable effects on the dynamic behavior and the minimum film thickness. Guyan reduction of the 2D pad elastic FE model was used for computational efficiency, where the pad's inner surface degrees of freedom are the master degrees of freedom and others become slaves.

Gadangi et al. [25] presented the bearing thermal effects on the bearing-shaft dynamic behavior under the sudden mass unbalance condition. The three different transient analyses are studied, which are (a) full time transient analysis, (b) linear analysis using dynamic coefficients, and (c) pseudo-time transient analysis considering static application of dynamic loads.

Fillon et al. [26] studied the pad thermal–elastic deformation effects on the TPJB dynamic behavior under unbalance loading. To evaluate the effective viscosity of the lubricant and pad thermal–elastic deformation, a pseudo-time transient analysis was developed. Both the temperature–viscosity variation and the operating film thickness due to the elastic–thermal pad deformations were found to have a strong influence on the bearing dynamic behavior of the bearing.

Monmousseau and Fillon [27] performed a nonlinear transient TEHD analysis for a TPJB under a dynamic loading condition. Both thermal and elastic deformations of the bearing pad were considered, and the numerical results were compared with experimental results. It was found that the bearing dynamic response was maximum around the critical speed. Chang et al. [28] predicted the static characteristics of a tilting pad journal bearing through TEHD analysis. They used a 3D FE model to solve the energy equation and the thermal deformation problem of the pad.

Sim and Kim [29] presented a THD numerical approach for flexure pivot tilting pad gas bearings accounting for a generalized Reynolds equation, 3D energy equation, and heat flux equations. Both rotor thermal and centrifugal expansions are considered.

Bang et al. [30] performed an experimental study to compare leading edge groove tilting pad journal bearings with conventional journal bearings under different running conditions. The power loss and pad temperature were measured with and without a seal tooth.

Tschoepe and Childs [31] measured the clearance both at room temperature and hot temperature immediately following tests to obtain cold and hot clearances. A 16%–25% decreased clearance was measured at hot clearance compared to room temperature. The thermal deformation of the shaft and bearing pads was measured and simulated.

Zhang et al. [32] presented redesigned structural parameters of a three-pad tilting-pad journal bearing with an 800 mm diameter used to support a 1150 MW nuclear power generator. The in-service bearing local melting damage was caused by a high temperature rise.

Suh and Palazzolo [3,4] separately predicted the thermal and elastic deformation of the bearing pad through their numerical model. From the point of view of the preload, the pad of the bearing was increased by both elastic and thermal deformation. Their study did not consider various driving conditions. If the bearing pad is thick enough, performance change due to elastic deformation of the bearing pad is generally not considered. Performance change due to thermal deformation of the bearing pad of tilting pad journal bearings has not been dealt with in most of the existing studies, and similarly is not considered when designing bearings. This is because, in order to predict the thermal deformation of the bearing pad, it is necessary to predict the viscous shear heat, which is the most dominant heat source in the bearing system, and to go through a complex process in which this heat transfer to the bearing structure must be considered.

Alakhramsing et al. [33] proposed a new mass-conserving Reynolds cavitation algorithm. The numerical analysis results were compared with the journal bearing experimental results. Mo et al. [34] performed a transient simulation of the journal bearing temperature using ANSYS Fluent. The bearing was used in the internal gear pump, where the complicated shaft motion is produced due to the time varying gear tooth contact load. A test rig was built for a comparison to the numerical results and validation. Nichols et al. [35] presented the effects of the lubricant supply flow rate on the bearing performance at a

steady state, and compared their experimental work with predicted data produced by software based on a thermo-elasto-hydrodynamic (TEHD) lubrication model. In this study, the thermal deformation of the bearing pad under the bearing operating conditions was predicted using a 3D FE model. In order to quantify the amount of deformation in terms of bearing performance change, the concepts of the performance thermal preload and geometric thermal preload were newly proposed, and the change in thermal preload under various conditions was predicted. Predicting the thermal deformation of the bearing pad from the viewpoint of the change in the preload has the advantage of predicting the performance change due to the thermal deformation of the pad under actual operating conditions compared to the initially designed bearing. This study aims to investigate performance changes due to thermal deformation of tilting pad journal bearings with various design variables and operating conditions in terms of preload change.

Arihara [36] predicted the performance of the tilting pad journal bearing using the TEHD lubrication model, and the results were compared with the experimental results. While the elastic deformation of the pad was predicted using the finite element method, an analytical model was used for the thermal deformation of the pad. The turbulence effect was considered using the turbulence viscosity model. Yang and Palazzolo [37,38] predicted the performance of tilting pad journal bearings using commercial software. In particular, the mixing of oil between the pads, which was the biggest weakness in the bearing analysis results based on the existing numerical model, was predicted using the 3D rans model. Chatterton et al. [39] predicted the performance of the tilting pad journal bearing when the pad cooled by various methods and compared it with the experimental results. Suh et al. [40] predicted the change of static and dynamic characteristics under various temperature boundary conditions around the tilting pad journal bearing.

If the viscosity drop due to viscous shear in the oil film is neglected, journal bearings with the same shape have the same dimensionless static and dynamic properties if they have the same Sommerfeld number. However, considering the thermal effect, the dimensionless characteristics may change even for bearings with the same Sommerfeld number. Jeung et al. [41] considered the change of dimensionless characteristics of tilting pad journal bearings of various sizes with the same Sommerfeld number. Bagiński et al. [42] experimentally analyzed the temperature change under various conditions for two gas foil bearings. Guo et al. [43] conducted a numerical analysis of journal bearings considering the asperity contact effect, viscosity–pressure effect as well as pad elastic deformation and thermal effects. Numerical results were compared with the experimental results.

Many studies have been conducted considering the thermal effect that occurs in journal bearings [3–43]. What these previous studies have in common is that the viscous shear heat generated in the oil film changes the lubricant viscosity, and the changed viscosity has a significant effect on the bearing performance. In addition, a few more advanced studies [3,4,37,38] predicted the thermal deformation of the bearing structure and the resulting change in oil film thickness using a 3D finite element model. However, there has been no study to correlate the specific thermal deformation shape of the bearing structure with the performance change of the bearing by quantifying it. Although Suh and Palazzolo [4] proposed the concept of thermal preload, it was not studied in terms of bearing performance change.

In this study, the thermal deformation shape of the pad of the tilting pad journal bearing was quantified in terms of the preload amount, which is one of the important design parameters of bearing. The change in the performance was predicted by quantifying the thermal deformation of the pad that occurs under various operating conditions in terms of the amount of preload.

## 2. Numerical Models

### 2.1. Lubrication Model

The oil film pressure was calculated using the variable viscosity Reynolds equation considering the viscosity change in the thickness direction of the oil film. A FE method was

used to discretize the target domain, and a triangular element was used. Equation (1) is the variable viscosity Reynolds equation, where $D_1$ and $D_2$ are defined in Equations (2) and (3), respectively. Here, $z$ is the film thickness direction and $\xi$ is an integral variable. The following assumptions are necessary in using the variable viscosity Reynolds equation.

(a)  Lubricant flow is full laminar.
(b)  Shaft curvature effect is neglected.
(c)  In the film thickness direction, the pressure is assumed to be constant.
(d)  Fluid inertia is not considered.
(e)  Fluid density is kept to be constant.
(f)  Incompressible Newtonian fluid.
(g)  At the slid and fluid interface, there is no slip.
(h)  Reynolds cavitation boundary condition.

Heat generation by viscous shear as well as convection and conduction must be considered in the oil film. The difference in lubricating oil velocity in the direction of thin oil film thickness generates viscous shear heat. $U$ denotes the journal linear velocity and $h$ is the film thickness.

Equation (4) shows the energy equation considering heat transfer and viscous shear. To discretize the problem domain, a 3D FE model with hexahedral elements was used. In Equation (4), $T$ represents the lubricant temperature, $\rho$ is the lubricant density, $c$ is the specific heat and $k$ is the heat conductivity. $x$, $y$, and $z$ are the circumferential, radial, and axial direction, respectively. $u$ and $w$ are the lubricant velocity in the $x$ and $z$ direction, respectively.

The Reynolds equation and the energy equation are combined with the relation between the temperature and viscosity of the lubricant as shown in Equation (5). The velocity field of the fluid calculated in the Reynolds equation is used to calculate the viscous shear heat in the energy equation, and the fluid temperature, which is the result of the energy equation, is used to calculate the fluid viscosity in Equation (5). The updated viscosity information is transferred to the Reynolds equation to evaluate the changed oil film pressure and velocity. Through this closed-loop calculation, the pressure and temperature within the oil film converge. In Equation (5), $\beta$ is a viscosity coefficient and $\mu_0$ is the reference viscosity at temperature $T_0$.

If heat transfer to the journal and pad surrounding the oil film is not taken into account, the steady-state analysis results of the bearing system can be obtained only with this simple convergence process. However, since this study considers heat transfer to the bearing pad and journal, and the resulting thermal deformation, more complex numerical models and algorithms are required. Details of the algorithm are covered in Section 2.3.

$$\nabla \cdot (D_1 \nabla p) + (\nabla D_2) \cdot U + \frac{\partial h}{\partial t} = 0 \tag{1}$$

$$D_1 = \int_0^h \int_0^y \frac{\xi}{\mu} d\xi dy - \frac{\int_0^h \frac{\xi}{\mu} d\xi}{\int_0^h \frac{1}{\mu} d\xi} \int_0^h \int_0^y \frac{1}{\mu} d\xi \, dy \tag{2}$$

$$D_2 = \frac{\int_0^h \int_0^y \frac{1}{\mu} d\xi \, dy}{\int_0^h \frac{1}{\mu} \, d\xi} \tag{3}$$

$$\rho c \left( u \frac{\partial T}{\partial x} + w \frac{\partial T}{\partial z} \right) = k \left( \frac{\partial^2 T}{\partial x^2} + \frac{\partial^2 T}{\partial y^2} + \frac{\partial^2 T}{\partial z^2} \right) + \mu \left[ \left( \frac{\partial u}{\partial y} \right)^2 + \left( \frac{\partial w}{\partial y} \right)^2 \right] \tag{4}$$

$$\mu = \mu_0 e^{-\beta(T - T_0)} \tag{5}$$

### 2.2. Heat Transfer and Thermal Deformation Model

The viscous shear heat is input to the spinning journal and bearing pad. Since the oil film and the pad meet in the same area, the general heat flux conditions were applied. On the other hand, since the journal rotates, the heat transferred to the journal was calculated using the average heat flux condition proposed by Gomiciaga and Keogh [44]. Since the journal is assumed not to whirl in the steady state condition, there is no circumferential temperature difference. The temperature at the interface should be constant in circumferential direction and the flowing heat flux should be also constant. The thermal boundary conditions between the oil film-journal and the oil film-pad are shown in Equations (6), (7), (8) and (9), respectively. The subscripts *L*, *B*, and *J* represent the lubricant, bearing and journal, respectively. *r* denotes the radial position, *R* is the radius of spinning journal, and *H* represents the thickness of thin film. $\theta$ denotes the circumferential position and $\omega$ is the rotor spin speed.

$$k_L \left. \frac{\partial T_L}{\partial r} \right|_{(r=R+H)} = k_B \left. \frac{\partial T_B}{\partial r} \right|_{(r=R+H)} \tag{6}$$

$$T_L|_{(r=R+H)} = T_B|_{(r=R+H)} \tag{7}$$

$$k_J \left. \frac{\partial T_J}{\partial r} \right|_{(\theta=0, r=R)} = k_L \left. \frac{\partial T_L}{\partial r} \right|_{(\theta=\omega t, r=R)} \tag{8}$$

$$T_J|_{(\theta=0, r=R)} = T_L|_{(\theta=\omega t, r=R)} \tag{9}$$

The heat transferred in this manner is transferred into the journal and bearing structure, and the temperature distribution in the structure can be calculated using the heat transfer equation. The calculated temperature distribution causes thermal deformation of the journal and pad. The thermal deformation of these structures changes the bearing oil film thickness. The amount of thermal expansion in the oil film direction of the bearing pad and journal was calculated, and this was reflected in the oil film thickness. Equation (10) shows this, where $h_{TEJ}$ denotes the journal radial thermal expansion of and $h_{TEP}$ is the radial thermal expansion of the pad. The corrected film thickness can be obtained by subtracting the amount of radial thermal expansion of the pad and journal from the thickness of the oil film.

$$
\begin{aligned}
h(\theta) = C_p &- \left( e_x - p_{pvt} \cos(\theta_p) \right) \cos(\theta) - \left( e_y - p_{pvt} \sin(\theta_p) \right) \sin(\theta) - (C_p - C_b) \cos(\theta - \theta_p) \\
&- \delta_{tilt} R_s \sin(\theta - \theta_p) - h_{TEJ}(\theta, z) - h_{TEP}(\theta, z)
\end{aligned} \tag{10}
$$

### 2.3. Algorithm

Figure 1 shows the algorithm for predicting the static equilibrium state of the bearing system. First, the thin film pressure and the fluid velocity are evaluated from the generalized Reynolds equation using the initial conditions. The oil film pressure is converted into forces and moments, and applied to the journal and pad. The fluid velocity is input into the energy equation to produce the oil film temperature, and the updated lubricant temperature changes the oil viscosity. In the first loop, the static positions of the bearing system converge. Static positions include journal radial position, pad tilt angle, and pivot elastic deformation. In the second loop, the oil film temperature converges with the bearing static positions.

Using the converged temperature and positions, heat flux boundary condition among spinning journal, oil film and pad is produced. Using these thermal boundary conditions, the temperature distribution of the spinning journal and pads, and the resulting thermal deformation is obtained.

The thermal deformation of such a structure is used in the generalized Reynolds equation by changing the oil film thickness, and the temperature distribution of the bearing structure is used as a thermal boundary condition in the energy equation to predict the oil film temperature.

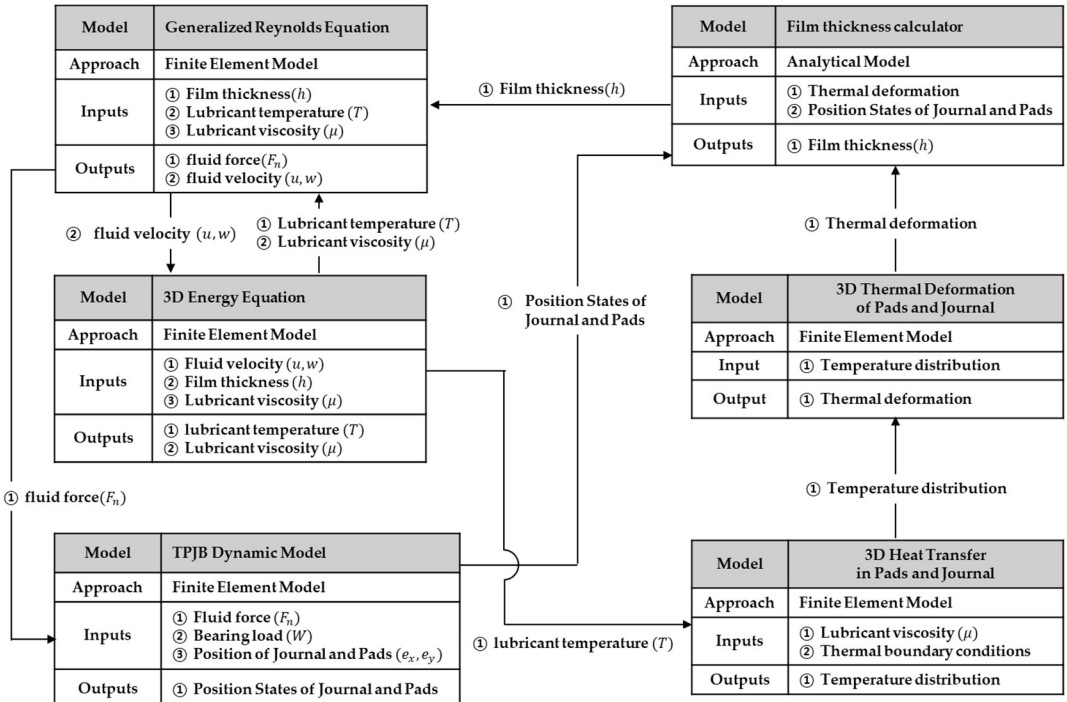

**Figure 1.** Algorithm for the analysis of the TPJB steady state with consideration of position, temperature and thermal deformation.

Finally, through these repeated calculations, the final convergence state can be determined by detecting that the bearing static positions, the oil film temperature, and the temperature change of the structure fall below a predefined criterion.

In this study, the calculation method of the bearing dynamic coefficient evaluated after steady-state prediction is not separately dealt with. This is because the method has already been covered in many existing studies [4,5,17], and most bearing researchers already have sufficient knowledge about it. In this study, a synchronously reduced dynamic coefficient was used.

*2.4. Suggestion of Thermal Preload*

2.4.1. Preload

One of the advantages of tilting pad bearings is that many design parameters such as the load direction, pivot offset, preload, and length-diameter (LD) ratio can be adjusted to meet the purpose of the bearing. Among them, the most important design parameter for bearing designers is the amount of pad preload. The preload is defined by the pad geometry rather than the actual load applied as in rolling element bearings.

Figure 2 represents an explanation of these preloads. When the center of the bearing pad and the journal coincide, the pad preload is defined as zero. If the pad has a positive preload, compared to the pad with zero preload, its shape is kept circular and wider. If it is further widened, the center of the pad circle and the center of the journal circle do not coincide. In Figure 2, if the geometric center of the journal is $O_b$ and the geometric center of the circular pad is $O_p$, the distance between $O_b$ and $O_p$ can be defined as $r$. At this time, the pad clearance as shown in Equation (11) and the bearing clearance, as given in Equation (12), are defined.

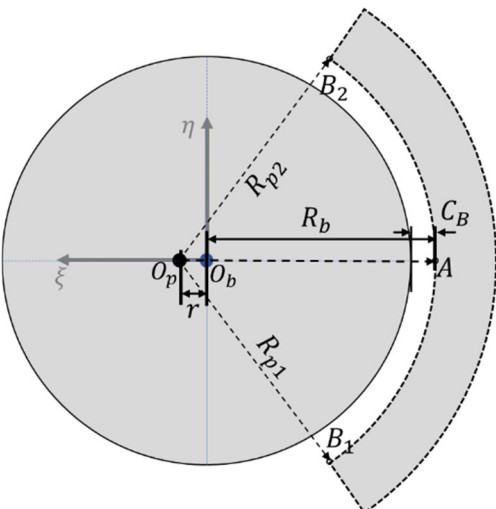

**Figure 2.** Journal bearing preload.

$R_b$ is defined as the length from the center of the journal to the inner surface of the pad at the pivot location, and $R_p$ is defined as the radius of the pad circle as shown in Figure 2. At this time, the preload is defined as Equation (13). A pad with a positive preload means that it is geometrically widened compared to a pad with zero preload, and a pad with a negative preload means that it shrinks more.

Typically, the preload has a value between 0 and 0.5. When the pad has a positive preload, a converging film section exists in the oil film even when the journal has zero eccentricity. This means that even if the journal has zero eccentricity, it can be supported by hydrodynamic forces due to the wedge effect.

$$C_P = R_p - R \tag{11}$$

$$C_B = R_b - R \tag{12}$$

$$M_P = 1 - \frac{C_b}{C_p} = \frac{r}{r + C_b} \text{ where } r = C_p - C_b \tag{13}$$

2.4.2. Performance Thermal Preload and Offset Thermal Preload

The viscous shear heat generated in the thin film is transmitted to the spinning journal and bearing pads, which are two structures in contact with the oil film, causing a temperature change and the resulting thermal deformation. The thermal deformation updates the oil film thickness and the pressure. Additionally, the temperature gradient of the bearing structure changes the lubricant temperature and oil film pressure. The numerical modeling method considering such a complex process is described in the previous section.

In particular, the pad thermal deformation causes a change in the shape of the bearing pad and thus a change in the preload. This change in preload may cause unexpected problems such as rotor vibration or deterioration of bearing performance.

In this study, the concept of thermal preload is introduced in order to quantify the thermal deformation caused by the temperature change of the bearing pad in terms of bearing performance change. Thermal preload can be divided into performance thermal preload and offset thermal preload. This section will define these, and the next section will investigate the change in the thermal preload and bearing performance under several operating conditions.

Figure 3a,b describe the performance thermal preload and offset thermal preload, respectively. The initial shape of the journal and pad without thermal deformation is drawn with a solid black line, and the deformed shape with thermal deformation is with a red dotted line.

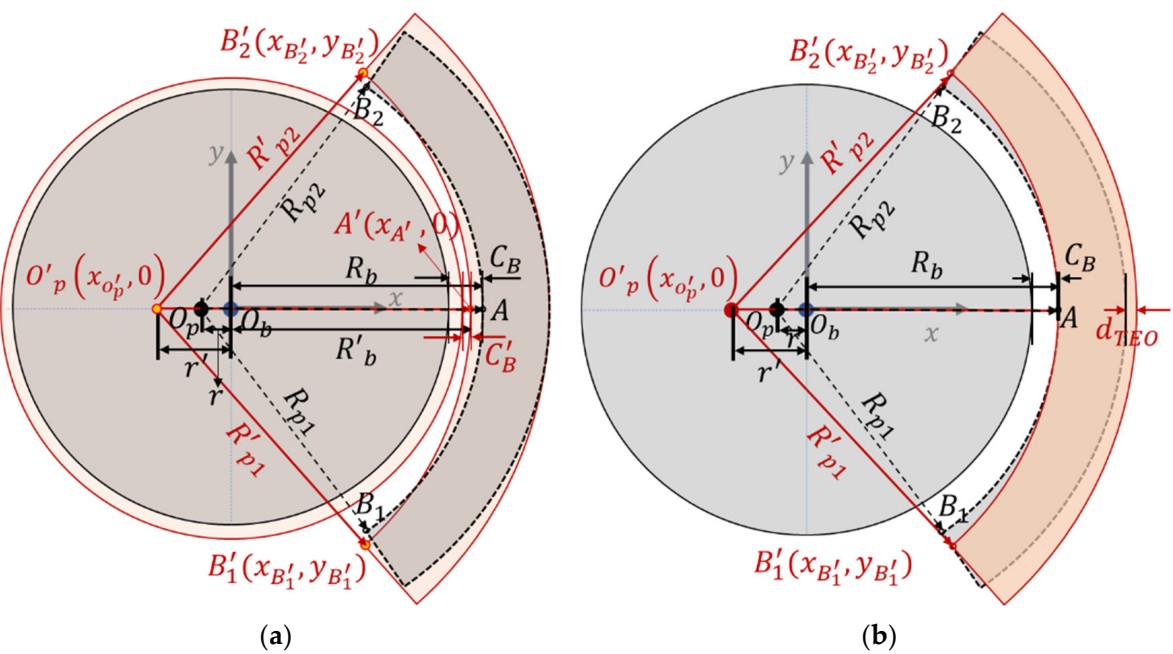

**Figure 3.** Schematic diagram for thermal preload change: (**a**) performance thermal preload; (**b**) offset thermal preload.

First, the performance thermal preload is defined. The performance thermal preload is defined in terms of the bearing performance and considers the thermal deformation of the spinning journal and the pads at the same time. Since the thermal deformation of the two structures changes the oil film thickness and the pressure, it is necessary to consider both from the viewpoint of bearing performance. Since the journal is spinning, there is no thermal gradient in the circumferential direction and only a thermal gradient in the radial direction. Hence, it is assumed that the journal shape after the thermal deformation maintains a perfect circle. The journal radius before the thermal deformation is $R_b$, and the radius after the deformation is $R_b'$. On the other hand, the pad has a thermal gradient in all directions, which causes complex deformation rather than simple expansion or contraction. To simplify this, it is assumed that the shape of the inner surface of the pad maintains a perfect circle after thermal deformation. As shown in Figure 3a, if the two points at both ends in the circumferential direction of the pad inner surface are $B_1$ and $B_2$, respectively, the positions of the two points will move to $B_1'$ and $B_2'$ after thermal deformation. Additionally, if the elastic deformation at the pivot is not considered, the position of the pad inner surface at the pivot location due to the thermal deformation moves from $A$ to $A'$. As shown in Figure 3a, the bearing clearance changes from $C_b$ to $C_b'$ due to thermal expansion of the pad and journal.

If the Cartesian coordinate system is located at the center of the journal, the position of each point mentioned above before and after thermal deformation can be expressed in the $xy$ coordinate system. The preload before the thermal deformation of the journal and pad is defined by Equation (13). Since the inner surface of the pad was assumed to maintain a perfect circle shape after thermal deformation and the center of the pad circle is moved to $O_p'$ after the thermal deformation, the $\overline{O_p'B_1'}$, $\overline{O_p'B_2'}$ and $\overline{O_p'A'}$ are identical. This relation is indicated in Equation (14). If Equation (14) is rearranged for $x_{O_p'}$, which is the position of $O_p'$ on the $x$-axis, Equation (15) is obtained. Eventually, the absolute value of $x_{O_p'}$ becomes $r'$, and the newly defined performance thermal preload is Equation (16).

$$\left( x_{B'} - x_{O_P'} \right)^2 + {y_{B'}}^2 = \left( x_{A'} - x_{O_P'} \right)^2 \tag{14}$$

$$x_{O'_P} = \frac{x_{A'}{}^2 - x_{B'}{}^2 - y_{B'}{}^2}{2(x_{A'} - x_{B'})} \tag{15}$$

$$M_{p'per} = 1 - \frac{C'_B}{C'_P} = \frac{r'}{r' + C'_B} = \frac{\left| x_{O'_P} \right|}{\left| x_{O'_P} \right| + C'_B} \tag{16}$$

If the performance thermal preload is defined in terms of the performance change of the bearing system due to system thermal deformation, the offset thermal preload is only related to the thermal deformation shape of the pad' inner surface. As shown in Figure 3b, only the thermal deformation shape of the pad inner surface is considered, while the journal thermal deformation is not considered. Unlike the performance thermal preload where the pad thermal expansion in the radial direction at the pivot location was taken into account where the point $A$ is moved to $A'$, the pad thermal deformation at the pivot location is offset by the amount of pad thermal deformation ($d_{TEO} = \overline{AA'}$) in the radial direction. This focuses only on the shape of the pad thermal deformation. Since the offset thermal preload also assumes that the pad maintains a perfect circle after the thermal deformation, the center of the pad circle after thermal deformation is $O'_p$. Since $\overline{O'_p B'_1}$, $\overline{O'_p B'_2}$ and $\overline{O'_p A}$ are identical as in the thermal performance preload, this relationship can be expressed as Equation (17). At this time, the position of point $A$ changed by thermal deformation is offset and the bearing clearance ($C_b$) does not change because the thermal deformation of the journal is not considered in the offset preload. If Equation (17) is arranged for $x_{o'_p}$, it becomes Equation (18), and the offset thermal preload is expressed as Equation (19).

In this section, both circumferential end points $B'_1$ and $B'_2$ of the pad were used to calculate the thermal preload. However, as will be explained in the following section, keep in mind that $B'_1$ and $B'_2$ can be moved to any position on the pad as well as the two ends for thermal preload calculation.

$$\left( x_{B'} - x_{O'_P} \right)^2 + y_{B'}{}^2 = \left( x_A - x_{O'_P} \right)^2 \tag{17}$$

$$x_{O'_P} = \frac{x_A{}^2 - x_{B'}{}^2 - y_{B'}{}^2}{2(x_A - x_{B'})} \tag{18}$$

$$M_{p'off} = 1 - \frac{C_B}{C'_P} = \frac{r'}{r' + C'_B} = \frac{\left| x_{O'_P} \right|}{\left| x_{O'_P} \right| + C_B} \tag{19}$$

## 3. Numerical Results and Discussions

### 3.1. Simulation Model

Tables 1 and 2 show the numerical model used for the prediction of thermal preload in this study. The bearing design parameters used in this study were adopted from those used in a previous study [41]. Details of where the bearing is used were not mentioned at the request of the bearing manufacturer. It has load on pad (LOP) configuration and the unit load is 1.41 MPa. According to the experience of the corresponding author of this study, it is expected that the unit load of 1.41 MPa can cause some performance change due to the pad elastic deformation. This performance change can vary greatly depending on the shape and thickness of the pad. Since this study deals with the performance change due to thermal deformation of the bearing pad, study on the performance change due to the pad elastic deformation is considered to be outside the scope of this study. Additionally, as the follow-up study to this study is the performance change due to the pad elastic deformation, it will be dealt with in detail in the next study.

**Table 1.** Lubricant and material properties [39].

| Lubricant | |
| --- | --- |
| Viscosity coefficient (Pa·s) | 0.0297 |
| Viscosity at 40 °C (N·s/m$^2$) | 0.0365 |
| Heat conductivity (W/(mK)) | 0.136 |
| Heat capacity (J/kg·°C) | 1886 |
| Density (kg/m$^3$) | 877 |
| **Journal** | |
| Young's Modulus (Pa) | $2.05 \times 10^{11}$ |
| Poisson's ratio | 0.3 |
| Reference temperature for thermal expansion (°C) | 25 |
| Thermal expansion coefficient (1/°C) | $1.22 \times 10^{-5}$ |
| Heat conductivity (W/(m °C)) | 42.6 |
| Heat capacity (J/(kg °C)) | 453.6 |
| **Pad** | |
| Young's Modulus (Pa) | $2.00 \times 10^{11}$ |
| Poisson's ratio | 0.3 |
| Reference temperature for thermal expansion (°C) | 25 |
| Thermal expansion coefficient (1/°C) | $1.21 \times 10^{-5}$ |
| Heat conductivity (W/(m °C)) | 51.9 |
| Heat capacity (J/(kg °C) | 453.6 |
| **Babbitt** | |
| Young's Modulus (Pa) | $5.3 \times 10^{10}$ |
| Poisson's ratio | 0.3 |
| Reference temperature for thermal expansion (°C) | 25 |
| Thermal expansion coefficient (1/°C) | $2.1 \times 10^{-5}$ |
| Heat conductivity (W/(m °C)) | 55 |
| Heat capacity (J/(kg °C) | 230 |
| **Housing** | |
| Young's Modulus (Pa) | $1.86 \times 10^{11}$ |
| Poisson's ratio | 0.3 |

The pad material is S25C and the journal is SCM440. The lining material is babbitt, and the lubricant is VG32. The lubricating supply temperature is kept at 50 °C, and convection boundary conditions are used around the pad and journal. The ambient temperature is kept constant at 30 °C and the convection coefficient is 100 W°C/m$^2$. A rocker back pivot is used. In Table 2, $D_p$ denotes the pivot diameter and $D_h$ denotes the housing diameter in contact with the pivot. The pivot stiffness is calculated using the Hertzian contact theory. The rotor spin speed is from 500 rpm to 1250 rpm.

Figure 4a describes the shape and operating conditions of the bearing system, and Figure 4b shows a cross-section of a 3D FE model for the heat transfer and thermal deformation calculation of the journal and pads. Figure 5 shows the element of the FE model in which the Reynolds equation is discretized.

FE models in this study are modeled in MATLAB R2022b, a numerical analysis software, and a parallel processing technique was used to improve the computation efficiency. The numerical model used in this study has already been verified in a previous study performed by the corresponding author of this paper [41].

**Table 2.** Bearing configuration and running conditions [39].

| Running Conditions | |
|---|---|
| Sommerfeld number | 0.162 |
| Rotor spin speed (rpm) | 500~1250 |
| Load direction | LOP |
| Ambient temperature (°C) | 30 |
| Convection coefficient (W°C/m²) | 100 |
| Lubricant supply temperature (°C) | 50 |
| Mixing coefficient at oil groove | 0.8 |
| **Bearing configurations** | |
| Number of pads | 5 |
| Pad arc length (°) | 60 |
| Offset | 0.5 |
| Preload | 0.4 |
| Unit load (MPa) | 1.41 |
| Bearing load (N) | 11,070 |
| Journal diameter (mm) | 105 |
| Pad thickness (mm) | 15 |
| Babbitt thickness (mm) | 1.5 |
| Pad clearance (mm) | 0.096 |
| Pad length (mm) | 75 |
| Pivot type | Rocker-back |
| $D_p$ (mm) | 114 |
| $D_h$ (mm) | 135 |

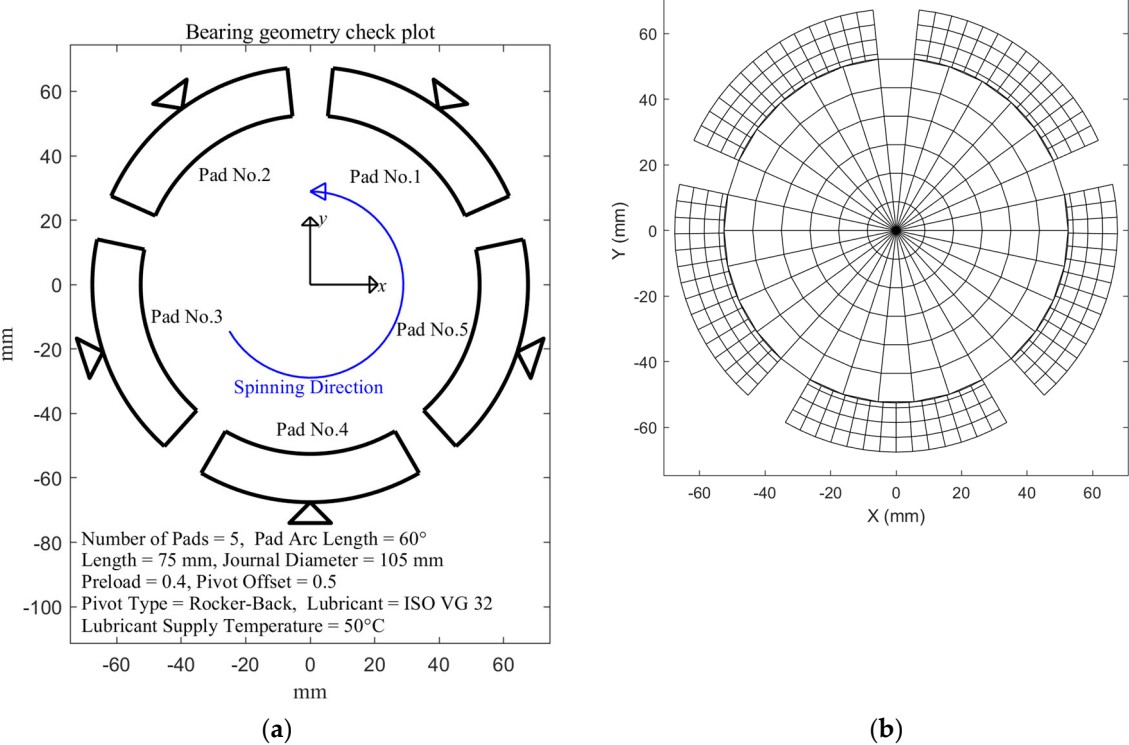

**Figure 4.** Bearing numerical model: (**a**) Geometry check plot. (**b**) Cross sectional view of 3D bearing FE model.

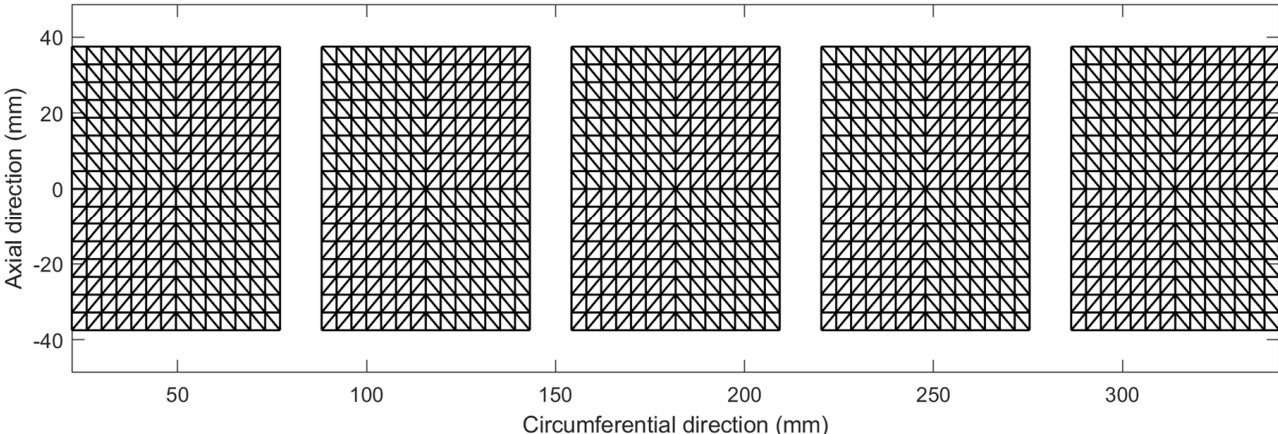

**Figure 5.** Fluid film FE model.

### 3.2. Pad Thermal Deformation Shape and Offset Thermal Preload

When the offset thermal preload is defined as given in Figure 3b of Section 2.4.2, both circumferential end points $B_1'$ or $B_2'$ of the pad were used. It is possible to define the thermal preload even by using the thermal deformation at any position in the pad as well as a point located at the circumferential end of the pad. In this case, the thermal preload can be calculated using the amount of thermal deformation at an arbitrary position of pad inner surface on the FE pad model. However, $B_1'$ and $B_2'$ cannot be located at point $A'$.

As the thermal preload was defined, it was assumed that the inner surface of the bearing pad remains perfectly circular after thermal deformation. In order to determine whether the shape of the pad after thermal deformation is a perfect circle, it is sufficient to check whether the offset thermal preload calculated at all nodes in the pad after thermal deformation is the same. If the offset thermal preload calculated at all nodes is the same, the pad inner surface maintains a circular shape even after thermal deformation. If the value is not constant, it may be determined that the pad is no longer circular due to the thermal deformation. In this section, the offset thermal preload after thermal deformation will be evaluated to confirm this assumption. This is because the offset thermal preload is only related to the thermal deformation shape of the bearing pad.

Figure 6 shows the distribution of the offset thermal preload for each pad according to the change in the rotor spin speed in the bearing numerical analysis model presented in Section 3.1. It can be seen that the thermal offset preload differs greatly depending on where $B_1'$ and $B_2'$ are located. When the spin speed is 500 rpm (see Figure 6a), the offset thermal preload change is within 0.16; however, when the rotor spin speed increases to 1250 rpm (see Figure 6f), the difference increases to more than 0.4.

Figure 7 shows the temperature distribution on the inner surface of the pad. Under normal operating conditions, where the temperature outside the pad is lower than the temperature of the lubricant heated by viscous shear, the maximum temperature of the pad exists on the inside surface of the pad. It can be seen that the maximum temperature increases as the spin speed increases. The temperature distribution on the inner surface of the pad is a good measure to check the integrity of bearing structures such as babbitts and lubricants; however, in this study, the thermal deformation of the structure should be additionally checked.

Figure 8 shows the amount of radial thermal deformation of the pad inner surface. It can be seen that the circumferential middle part of the pad swells into the bearing and the edge of the pad deforms outward. Through this result, it can be seen that the pad is thermally deformed in the unfolded form. Figure 9 shows the temperature distribution of the axial central section of the bearing system composed of the journal, lubricant and pads. The circumferential temperature distribution of the pad is similar to the oil film temperature distribution. In addition, since the ambient temperature outside the pad of the bearing system in this study is kept at 30 °C, a temperature gradient in the radial direction

is also observed within the pad. However, it is difficult to obtain quantitative information on performance change due to temperature gradient and thermal deformation using only these data.

Figure 10 shows only the axial central value of the thermal preload in the plane drawn in Figure 6. It can be seen that the difference between the maximum and minimum values of the preload distribution increases as the rotor spin speed increases. Compared to the bearing design preload of 0.4, it can be seen that the preload changes due to the thermal deformation of the pad. Additionally, as the pad temperature increases, the difference between the thermal preload near the center of the pad and outside the pad increases.

Figures 7 and 9 show that at 500 rpm, the maximum temperature of the lubricant and pad is distributed near the circumferential center of the pad; however, as the spin speed increases, the maximum temperature moves counterclockwise. The large change in preload within the same pad observed in Figure 10 appears to be related to this.

Although it has been confirmed that the pad does not maintain a perfect circular shape after thermal deformation, it is expected that the effect of pad thermal deformation on the bearing performance change can be quantitatively determined using the newly defined thermal preload in this study. In particular, in the case of the preload measured near the pivot location (the circumferential center of the pad), the difference between the values measured on the left and right around the pivot is large. This is because, as shown in Figures 7–9, the position of the maximum temperature on the pad and the resulting maximum thermal expansion changes near the pivot location.

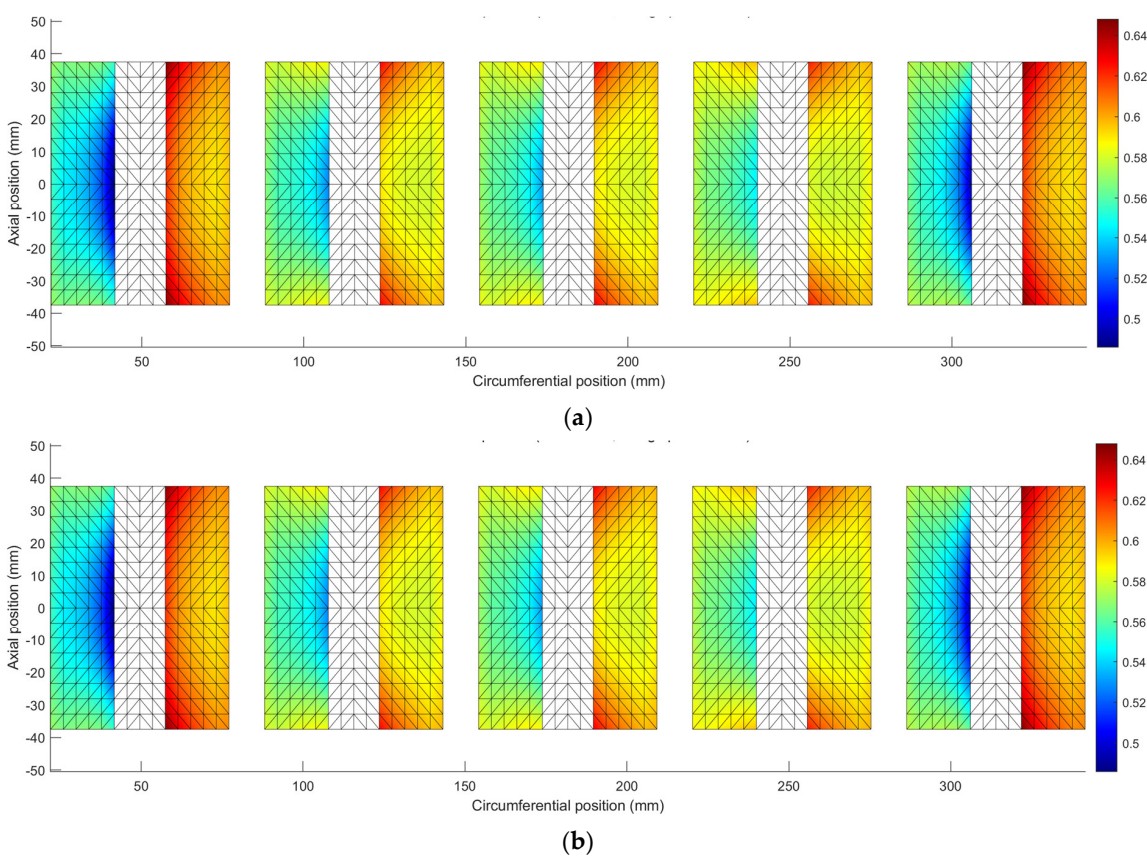

**Figure 6.** *Cont.*

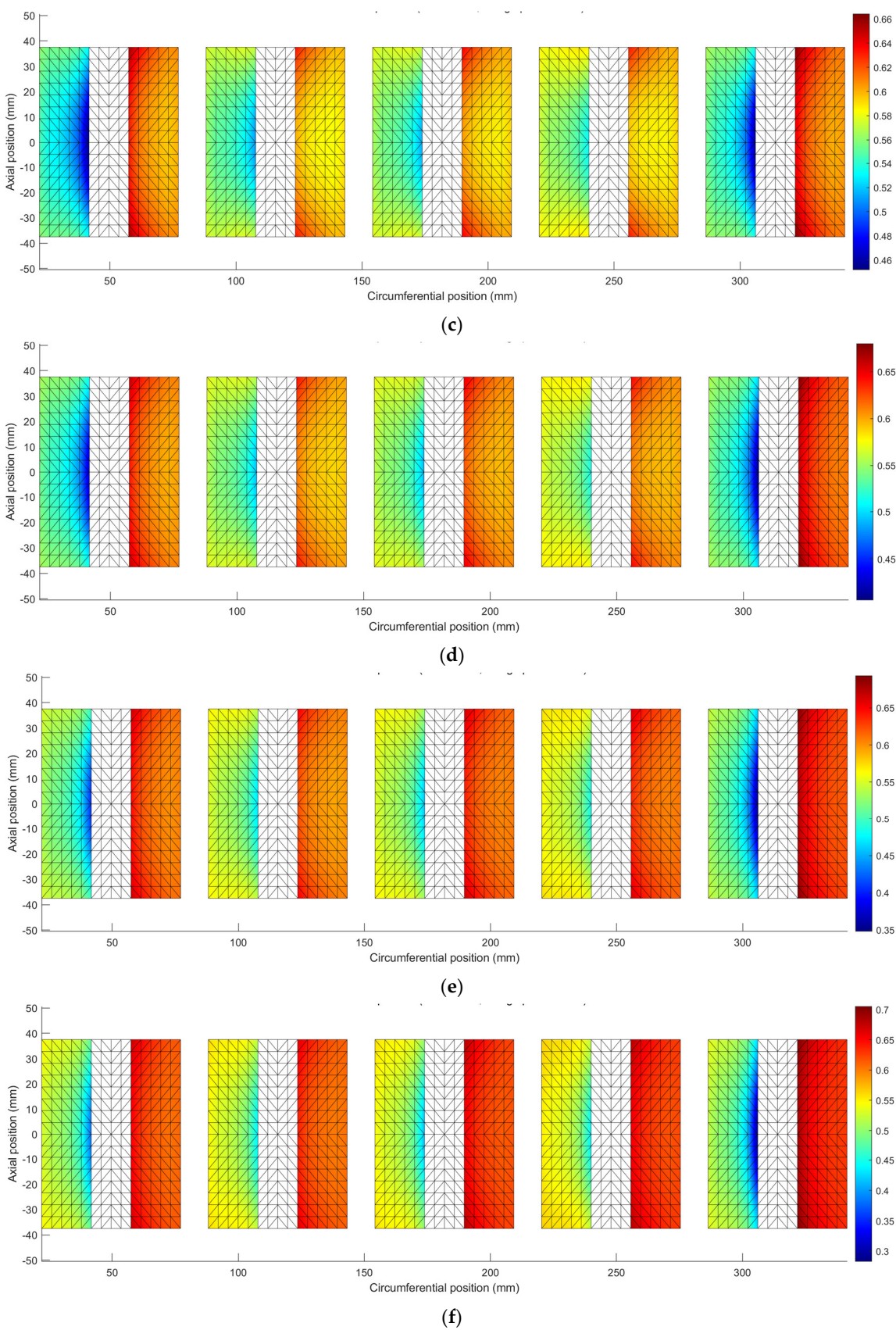

**Figure 6.** Offset thermal preload: (**a**) 500 rpm; (**b**) 750 rpm; (**c**) 1000 rpm; (**d**) 1250 rpm; (**e**) 1500 rpm; and (**f**) 1750 rpm.

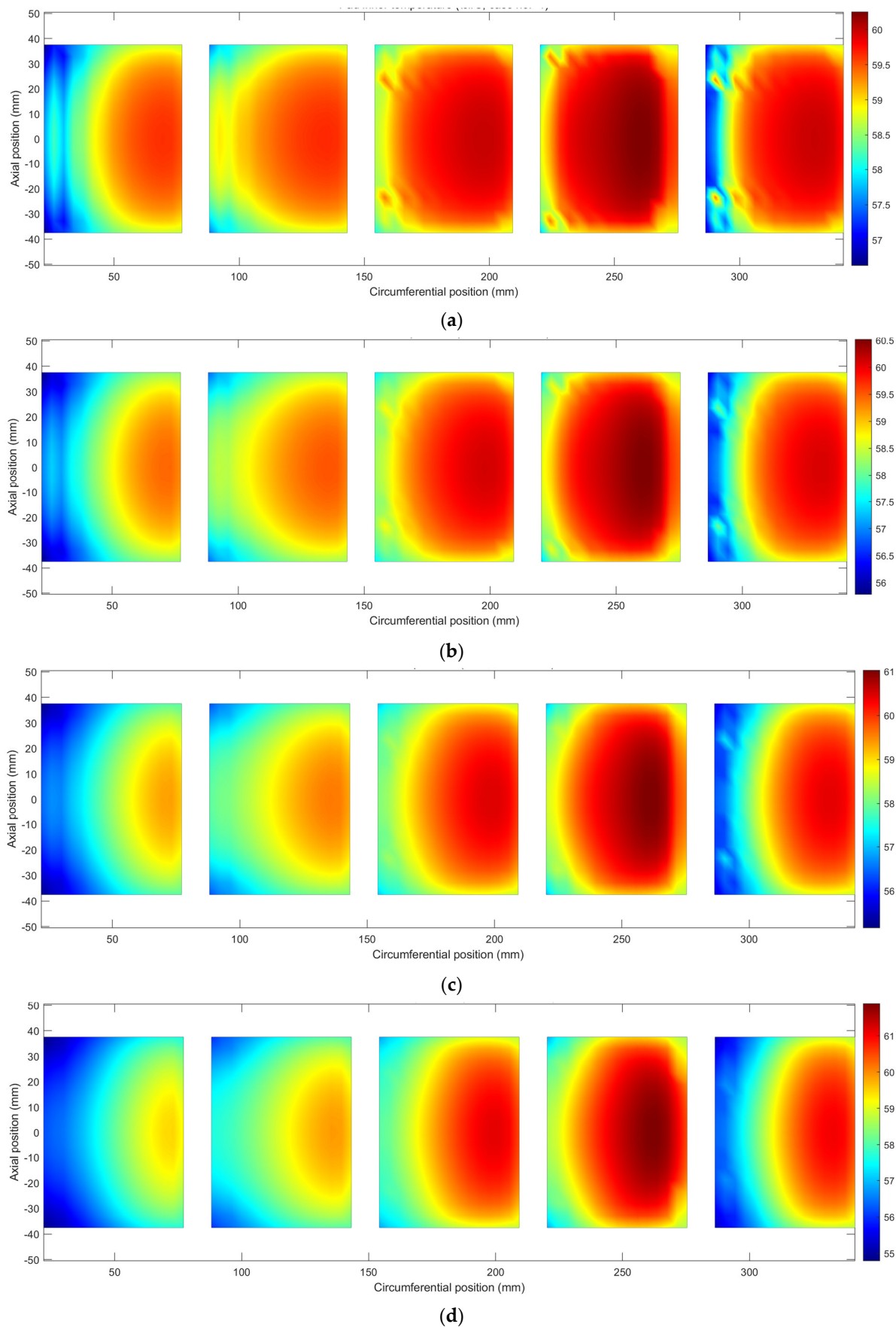

**Figure 7.** *Cont.*

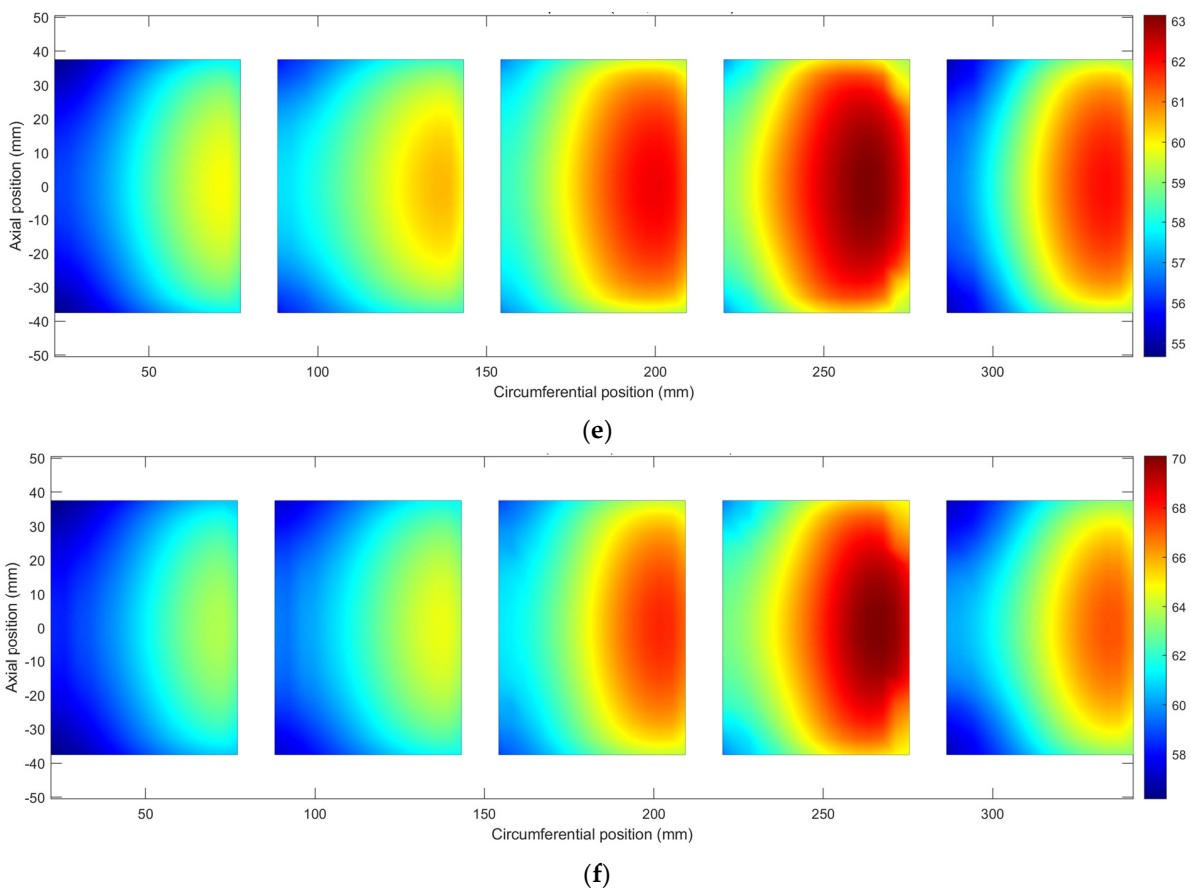

**Figure 7.** Temperature distribution on pad inner surface: (**a**) 500 rpm; (**b**) 750 rpm; (**c**) 1000 rpm; (**d**) 1250 rpm; (**e**) 1500 rpm; and (**f**) 1750 rpm.

As seen in Figure 8, the maximum radial thermal deformation of the bearing pad is less than 1% of the bearing clearance ($C_b$). This is an error that can be sufficiently generated during bearing pad machining, and such small thermal deformation may not significantly affect bearing performance. In the next section, further investigation will be conducted on the effect of pad thermal deformation on bearing performance.

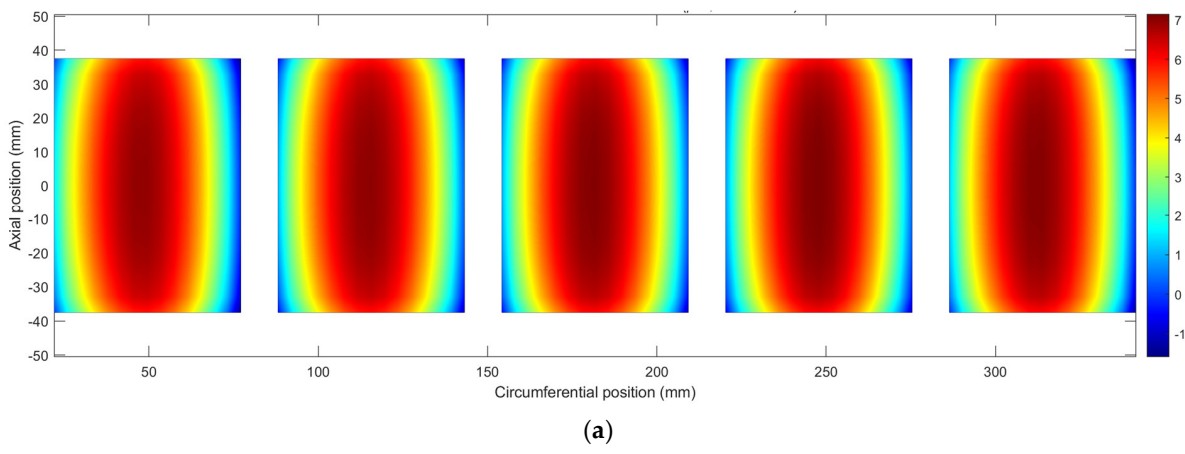

**Figure 8.** *Cont.*

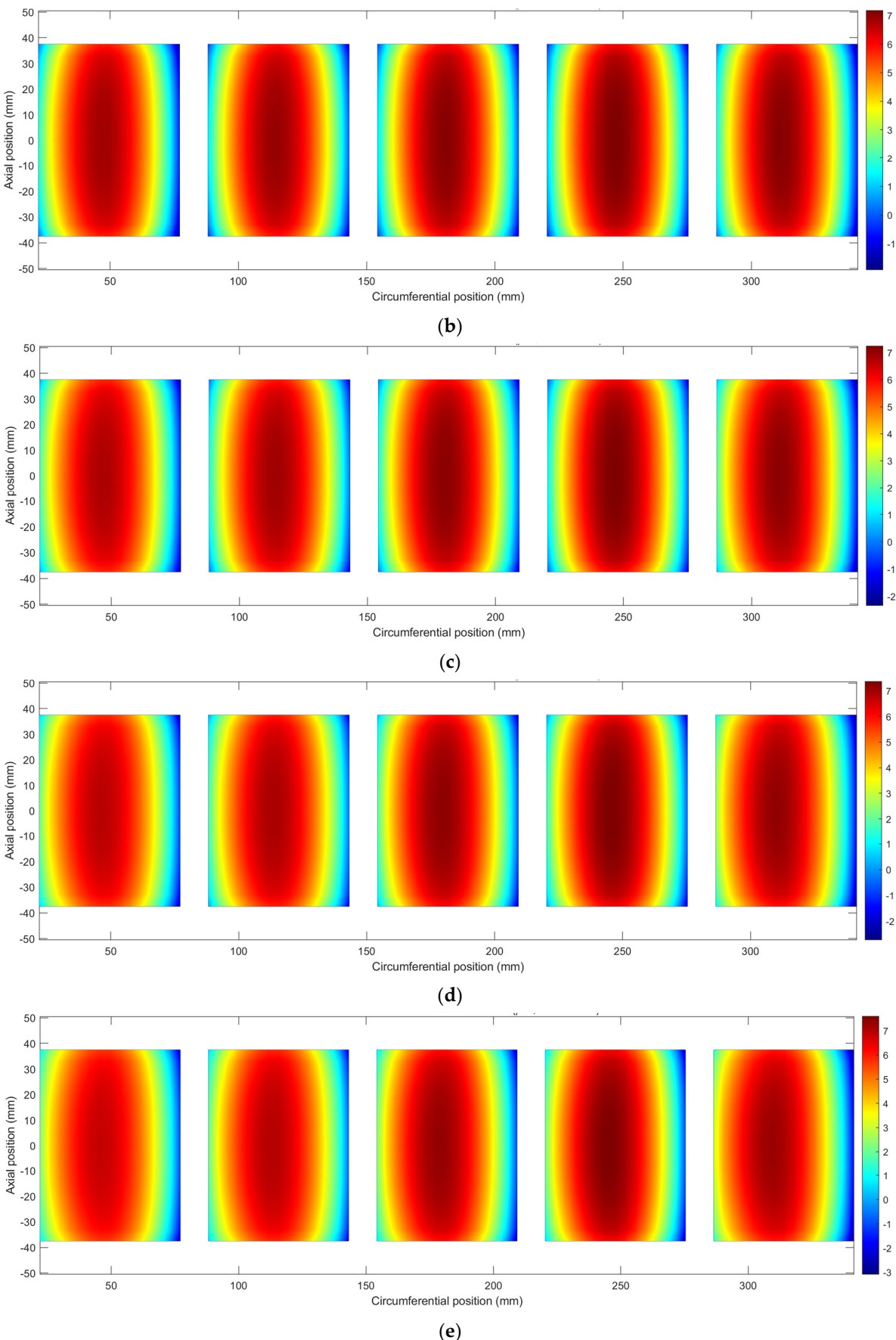

**Figure 8.** *Cont.*

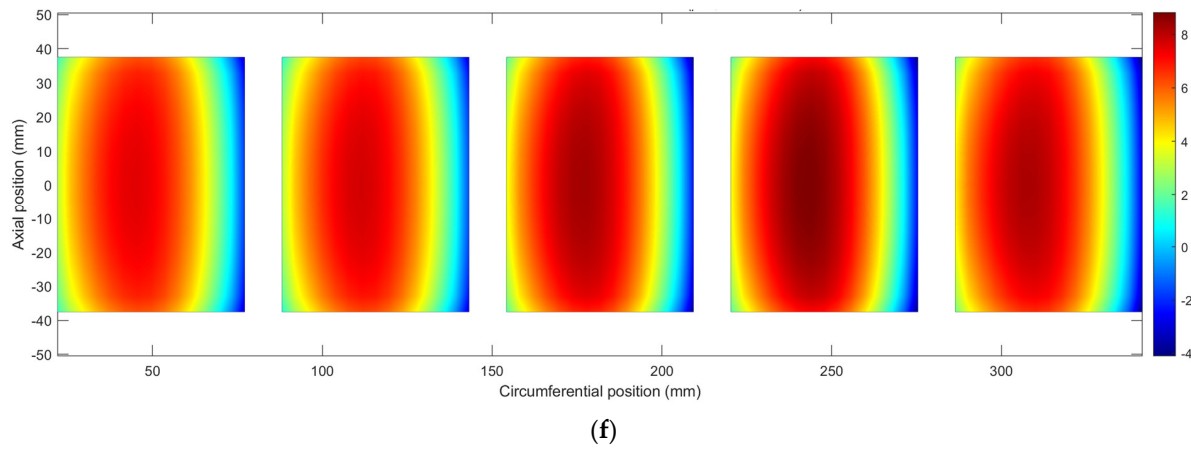

(**f**)

**Figure 8.** Pad thermal deformation (μm): (**a**) 500 rpm; (**b**) 750 rpm; (**c**) 1000 rpm; (**d**) 1250 rpm; (**e**) 1500 rpm; and (**f**) 1750 rpm.

(**a**)

(**b**)

(**c**)

(**d**)

**Figure 9.** *Cont.*

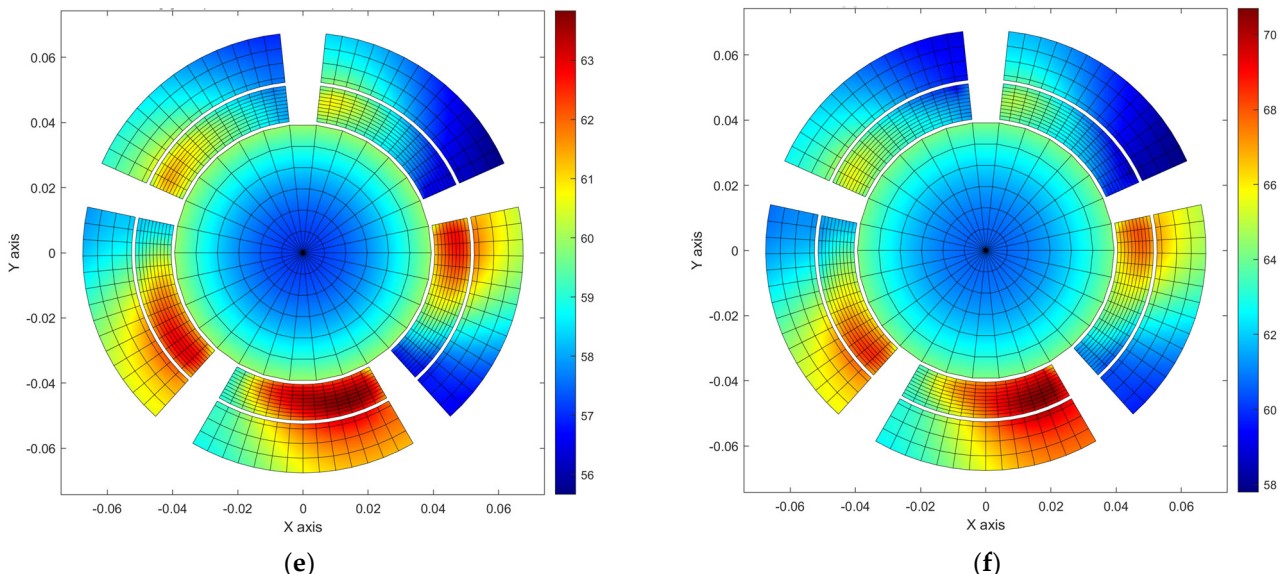

(**e**)     (**f**)

**Figure 9.** Temperature distribution of bearing system at axial center: (**a**) 500 rpm; (**b**) 750 rpm; (**c**) 1000 rpm; (**d**) 1250 rp€(**e**) 1500 rpm; and (**f**) 1750 rpm.

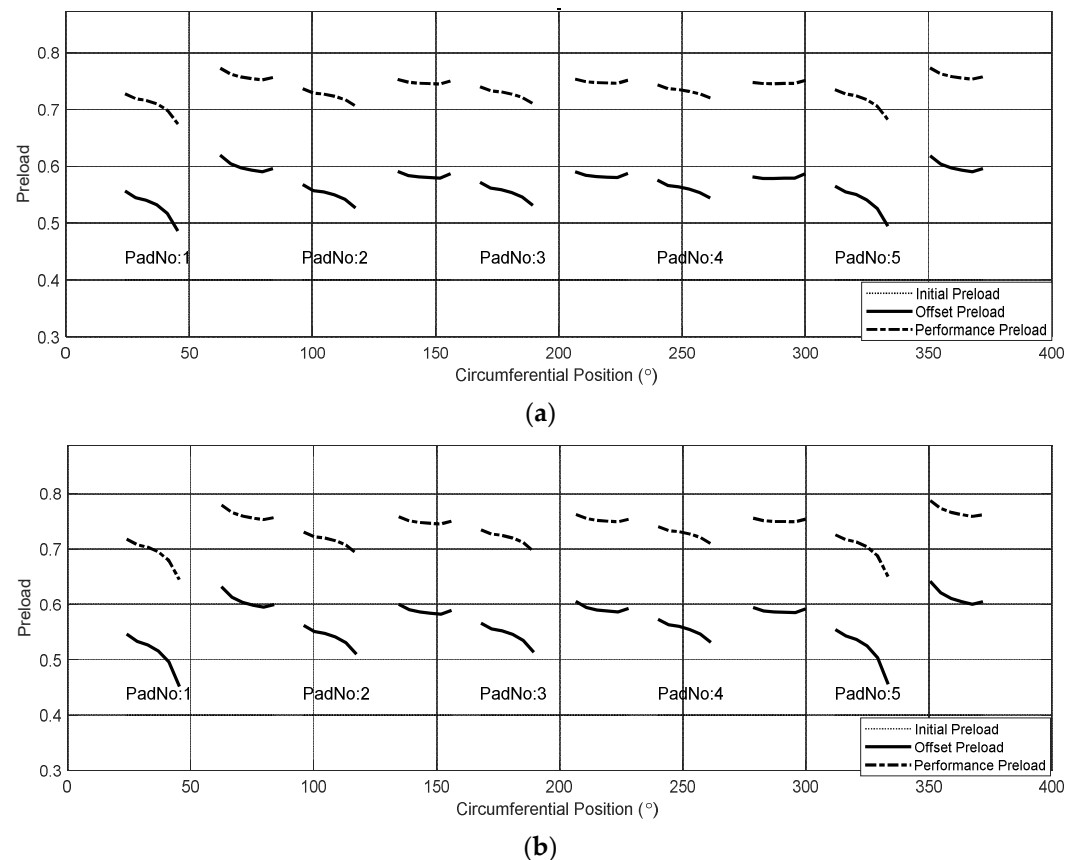

(**a**)

(**b**)

**Figure 10.** *Cont.*

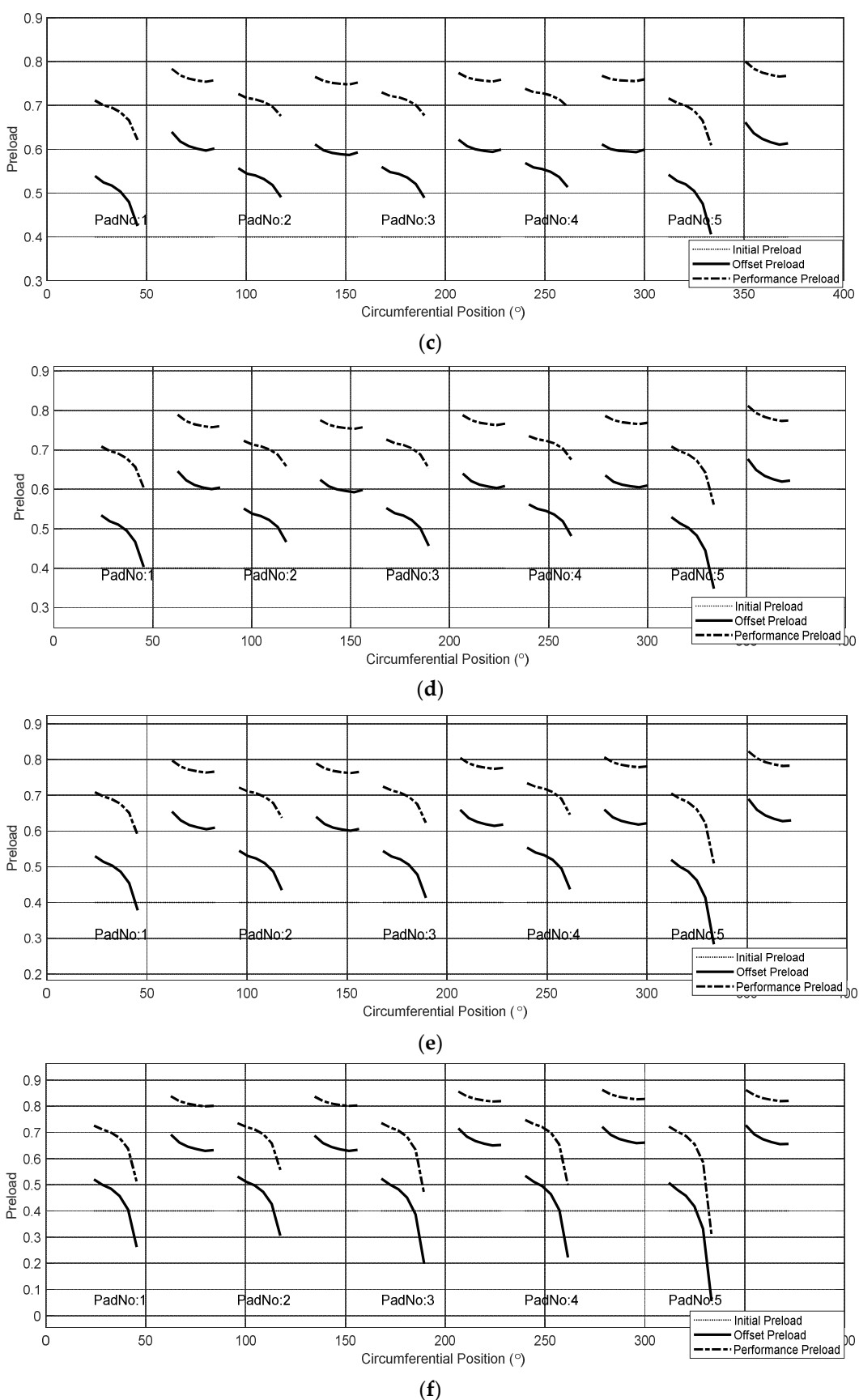

**Figure 10.** Thermal preload at axial center: (**a**) 500 rpm; (**b**) 750 rpm; (**c**) 1000 rpm; (**d**) 1250 rpm; (**e**) 1500 rpm; and (**f**) 1750 rpm.

### 3.3. Thermal Deformation Effect

In this section, the effect of pad thermal deformation on the bearing performance will be investigated. To this end, a performance analysis is additionally performed on a comparative model in which the thermal expansion coefficient of the pad was set to zero. All other characteristics of the model with a zero thermal expansion coefficient of the pad are identical to those of the bearing model presented in Section 3.1. As confirmed in the previous section, the maximum magnitude of the radial thermal deformation of the pad was less than 1% of the bearing clearance ($C_b$). We will take a closer look at how these small pad thermal deformations affect bearing performance.

In Section 3.2, it can be seen that the difference between the maximum thermal preload and the minimum thermal preload within the same pad increases as the rotor spin speed and pad temperature increase. To find out whether the change in bearing performance is correlated with either the central thermal preload or the outer thermal preload, the average of the thermal preloads measured at both ends of pad no.4, which is in the load direction, the mean of the thermal preload measured at the innermost part of the circumferential direction of pad no.4 is shown in Figure 11.

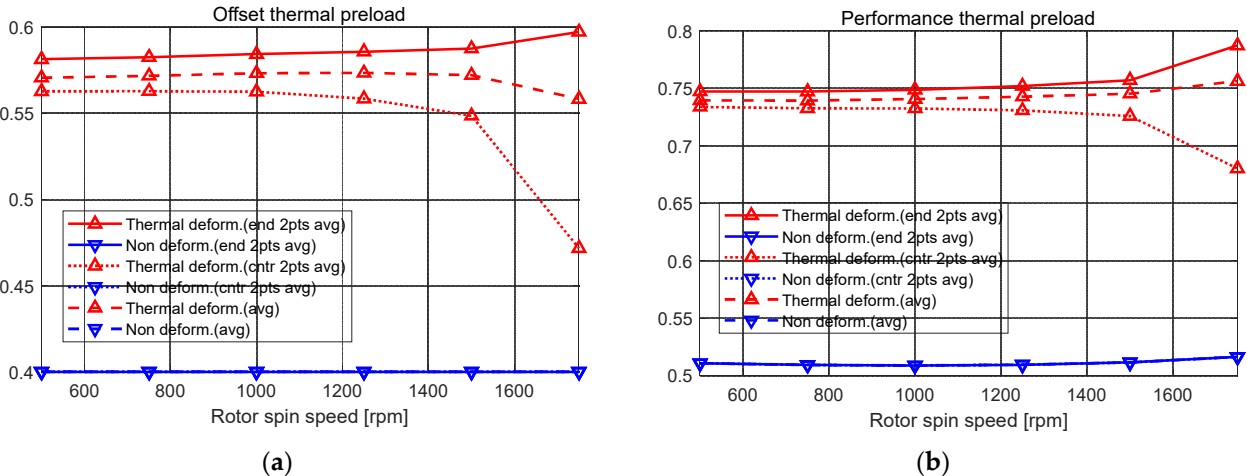

**Figure 11.** Thermal preload change of bottom pad considering two circumferential end points at axial center: (**a**) Offset thermal preload, and (**b**) performance thermal preload.

Figure 11a,b show the offset thermal preload and performance thermal preload, respectively. The blue line shows the thermal preload of the model in which the thermal expansion coefficient of the pad is zero, that is, the thermal deformation of the pad is not considered. The red line indicates the thermal preload in which the thermal deformation is considered. In the legend of the result plot, 'end 2pts avg' is the average value measured using both ends of the pad ($B_1'$ and $B_2'$ in Figure 3b). On the other hand, 'cntr 2pts avg' represents the average value of the thermal preload measured by moving $B_1'$ and $B_2'$ to the two closest nodes from the pivot. 'avg' represents the average value of the total offset thermal preload.

In Figure 11a, it can be seen that there is no change in the thermal preload of the model without the pad thermal deformation. This is because the offset thermal preload represents a change in the shape of the inner surface of the pad.

It can be seen that the thermal preload with thermal deformation using the two ends increases as the pad temperature and rotor spin speed increase. This is because the pad is thermally deformed into an unfolded shape with an increase in the pad temperature. On the other hand, it can be seen that the thermal preload using two points inside the pad decreases as the temperature of the pad increases. This is because the pad swells inward due to the maximum temperature that exists near the center of the pad. On the other hand, the offset thermal preload averaging all points has an intermediate value between the other two values.

Figure 11b shows the performance thermal preload. The performance thermal preload of the model without thermal deformation shows a value greater than 0.5 compared to the design parameter of 0.4 since the journal thermally expands even if the pad is not thermally deformed. The performance thermal preload has a similar tendency compared to the offset thermal preload.

Figure 12 shows the change in bearing static performance with the journal spin speed. Eccentricity is smaller when the pad thermal deformation is considered. Figure 12c shows the average temperature and the maximum temperature change. It can be seen that the temperature is high when the thermal deformation of the pad is considered. Figure 13d shows the power loss. When thermal deformation of the pad is considered, the power loss is high, which is due to the thinner oil film thickness due to the pad thermal expansion.

Figure 13 shows the bearing dynamic performances change according to the rotor spin speed. In the case of the cross coupled terms ($k_{xy}$, $k_{yx}$, $c_{xy}$, $c_{yx}$), they are not provided since they are close to zero. In the case of the direct stiffness terms ($k_{xx}$, $k_{yy}$, see Figure 13a,b), when the pad thermal deformation is considered, the value is higher than that of the other case. The inward swelling of the pads causes a decrease of the film thickness and the resultant decrease in the 'cntr 2pts avg' preload, which is the same as decreasing the preload to secure high rigidity in the general bearing design stage.

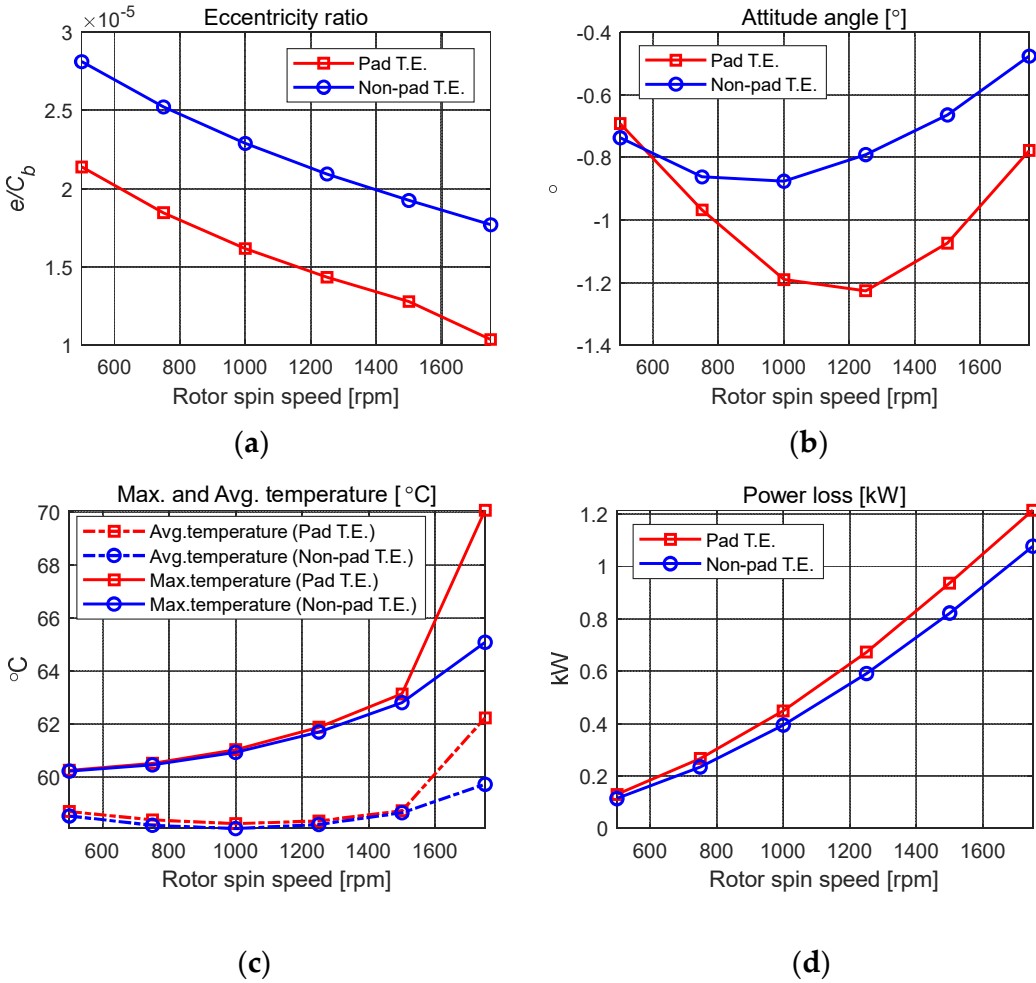

**Figure 12.** Bearing static properties: (**a**) Eccentricity, (**b**) attitude angle, (**c**) maximum temperature in bearing pads, and (**d**) power loss.

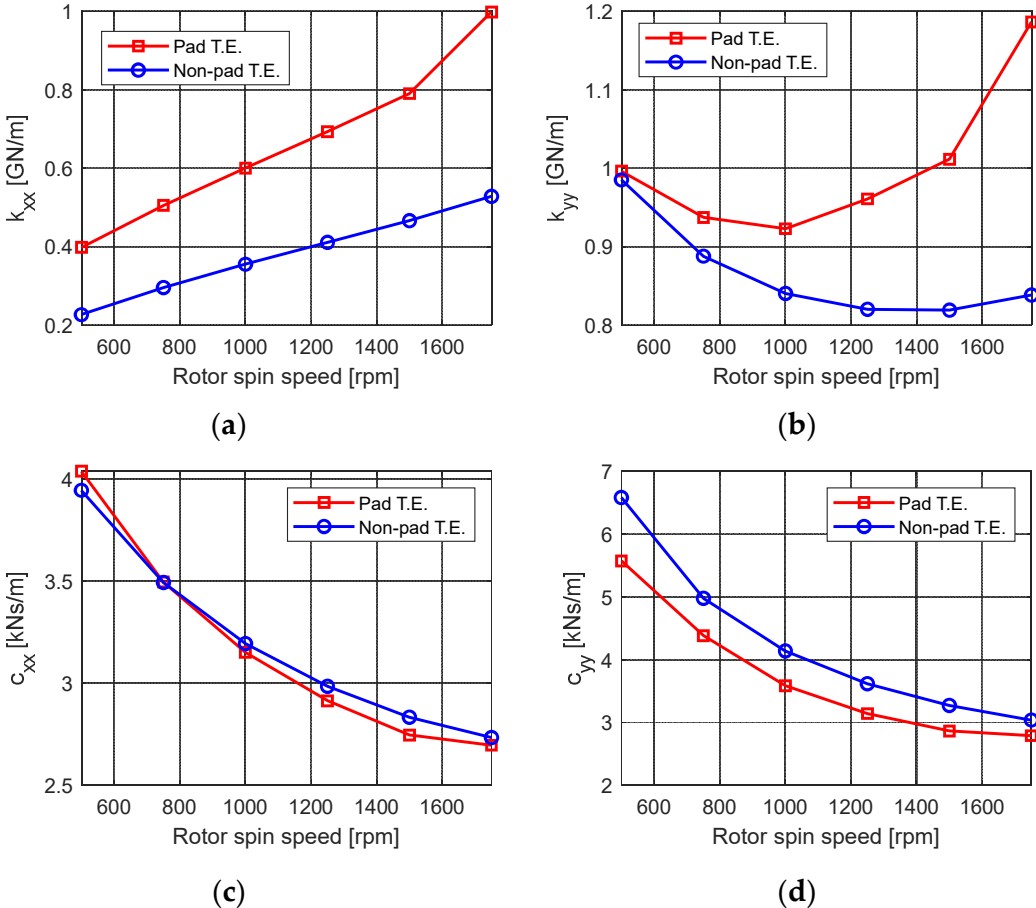

**Figure 13.** Bearing dynamic properties: (**a**) $k_{xx}$, (**b**) $k_{yy}$, (**c**) $c_{xx}$, and (**d**) $c_{yy}$.

In the case of the direct damping terms $c_{yy}$ (see Figure 13d), when the bearing thermal deformation is considered, the value is lower than that of the case without the bearing thermal deformation. On the other hand, it can be seen that the stiffness coefficient $c_{yy}$ in the load direction is not affected much by the thermal deformation of the pad compared to $c_{xx}$. In general, the bearing damping coefficient tends to decrease as the preload increases, indicating that it is related to the 'end 2pts avg' preload.

Comparing the performance change with and without thermal deformation of the bearing pad is not simple. It cannot be guaranteed that these differences in bearing performance will show the same trend in different bearings. However, the pad thermal expansion reduces the oil film thickness, and the reduced film thickness changes the fluid pressure and temperature. The changed oil film pressure changes the static and dynamic characteristics of the bearing, and the changed oil film temperature changes the viscosity and pressure of the oil film, and a complicated process of changing the thermal deformation of the bearing pad again occurs. The important aspect is that it may be difficult to accurately predict the actual bearing performance if the thermal deformation of the bearing pad is not taken into account. In particular, it should be noted that this difference tends to increase as the temperature change increases.

## 4. Conclusions

In tilting pad journal bearings, a temperature gradient in the bearing pad and thermal deformation occur at the same time due to the viscous heat transfer in the oil film. In this study, to investigate the effect of the thermal deformation of the pad on the bearing performances, the temperature gradient and thermal deformation within the pad were evaluated using a 3D FE model. The innovation points of this study compared to previous studies are as follows.

(a) The concept of the offset thermal preload was suggested to quantify the thermal deformation shape of the bearing pad.
(b) The concept of the performance thermal preload was suggested to quantify the thermal deformation of the bearing structure and the resulting bearing performance change.
(c) It was found that as the temperature of the bearing increases, the performance change of the bearing due to the thermal deformation of the pad also increases.

The preload of tilting pad journal bearings is one of the critical design parameters that bearing designers consider to achieve their goals in the design stage. However, although it can be predicted that the preload changes during bearing operation, there has been no study to predict this quantitatively. This study suggests that the offset thermal preload and the performance thermal preload are other important performance parameters that bearing designers should consider. The important findings from this study are as follows.

(a) The original shape of the bearing pad does not maintain its original circular shape due to the thermal deformation.
(b) The 'end 2pts avg' preload increases due to the pad thermal deformation, which decreases the direct damping terms, while the 'cntr 2pts avg' decreases due to the pad thermal deformation, which increases the direct stiffness terms.
(c) The model neglecting the pad thermal deformation showed a stiffness coefficient up 92% higher than that of the model not considering the thermal deformation.
(d) When the pad thermal deformation was considered, the damping coefficient was up to 10% higher than when it was not taken into account; however, the effect seems insignificant compared to the increase in the stiffness coefficient.

Through these research results, it can be seen that the bearing designer should consider the accurate prediction of the viscous shear heat generated in the oil film and the change in performance due to the thermal deformation of the pad during the design stage.

**Author Contributions:** Conceptualization, J.S. and Y.-D.C.; methodology, Y.-D.C.; software, J.L. (Jiyoung Lee) and J.S.; validation, J.L. (Jiyoung Lee); formal analysis, J.L. (Jiheon Lee); investigation, J.L. (Jiheon Lee); resources, J.L. (Jiheon Lee); data curation, Y.-D.C.; writing—original draft preparation, J.S. and Y.-D.C.; writing—review and editing, J.S.; visualization, J.S.; supervision, Y.-D.C.; project administration, Y.-D.C.; funding acquisition, Y.-D.C. All authors have read and agreed to the published version of the manuscript.

**Funding:** This research was supported by Nano·Material Technology Development Program through the National Research Foundation of Korea(NRF) funded by Ministry of Science and ICT (No. 2020M3H4A3106186).

**Data Availability Statement:** Not applicable.

**Conflicts of Interest:** The authors declare no conflict of interest.

## Nomenclature

The majority of symbols and notations used throughout the paper are defined below for quick reference. Others are clarified with their appearance in case of need.

| | |
|---|---|
| $p$ | Pressure |
| $U$ | Lubricant velocity |
| $u$ | Lubricant velocity in $x$ direction |
| $w$ | Lubricant velocity in $z$ direction |
| $\rho$ | Density |
| $c$ | Heat capacity |
| $t$ | Time |
| $k$ | Heat conductivity |
| $x$ | $x$ position in cartesian coordinate |
| $y$ | $y$ position in cartesian coordinate |

|   |   |
|---|---|
| $z$ | $z$ position in cartesian coordinate |
| $r$ | Radial position |
| $\theta$ | Circumferential position |
| $\mu$ | Viscosity |
| $\mu_0$ | Reference viscosity for lubricant viscosity change due to temperature |
| $T$ | Temperature |
| $T_0$ | Reference temperature for lubricant viscosity change due to temperature |
| $C_P$ | Pad clearance |
| $C_B$ | Bearing clearance |
| $R_p$ | Pad radius |
| $R_b$ | Bearing radius |
| $R_s$ | Shaft radius |
| $R$ | Journal radius |
| $e_x$ | Journal position in $x$ direction |
| $e_y$ | Journal position in $y$ direction |
| $\theta_p$ | Pivot circumferential position |
| $\delta_{tilt}$ | Pad tilt angle |
| $M_p$ | Preload |
| $h$ | Film thickness |
| $h_{TEJ}$ | Film thickness changes due to journal thermal deformation |
| $h_{TEP}$ | Film thickness changes due to pad thermal deformation |

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
