# Peer review of "Thermal Preload for Predicting Performance Change Due to Pad Thermal Deformation of Tilting Pad Journal Bearing"

_lubricants, doi:10.3390/lubricants11010003_

Round 1

Reviewer 1 Report

Great quality the study to investigate the quantitative change in performance due to thermal deformation of the swing bearing in terms of the change in the amount of preload. However, the article needs some modifications.

1.     The first three paragraphs of the introduction do not contain any references. Authors need to cite the origins of these claims.

2.     Is there real data for this study so that it can be compared with the simulated ones?

3.     As the work was carried out entirely by simulation, the title should contain this information.

Author Response

Dear Reviewer,

All authors of this paper are grateful to the reviewer for the suggestions and comments.

We are responding as follows to the opinions of the reviewers.

  1. The first three paragraphs of the introduction do not contain any references. Authors need to cite the origins of these claims.
  • As shown in the revised paper, new references are added to the first three paragraphs
  1. Is there real data for this study so that it can be compared with the simulated ones?
  • This is an extension of the former research[41]; the original bearing numerical model was verified by being compared with a reference. This is newly mentioned in the line number 399 as following.
  • ‘The numerical model used in this study has already been verified in previous study performed by the corresponding author of this paper [41].’
  1. As the work was carried out entirely by simulation, the title should contain this information.
  • The authors really appreciate the reviewer’s advice. Since this study is all related to numerical analysis, the authors tried to change the title to reflect this, but we have not been able to find a title that could express the contents of this study more clearly. If there is a title the reviewer can suggest, we will actively reflect it We sincerely appreciate it. 

Reviewer 2 Report

The manuscript offers no new knowledge than that other authors have offered years (decades) ago. Note that current predictive models include both thermal crowning and mechanical pressure deformation. Accounting for one effect and dismissing the other is NOT sound. Palazzolo and students, Hagemann and co-workers, San Andres and students, Fillon and students, etc. and my others have tackled the problem. The current work is a step back rather than a step forward. Note that mechanical (pressure) deformations affects the pivot deflection besides the pad. 

The authors must reduce the literature review to showcase the papers relevant to the current study and not provide a list of prior work with little critique of past undertakings. After the length analysis and conclusions, the conclusion section does not quantify findings but only qualifies them.  The manuscript presents not a single comparison to past data, in particular recent data from Childs et al, San Andres et al, Hagemann et al. 

Author Response

Dear Reviewer,

The concept of thermal preload proposed in this study, together with the concept of elastic preload to be proposed by this author in the future, will present a specific guide for performance changes that designers should consider in the design stage of tilting pad journal bearings.

Also, Dr Suh, the corresponding author of this paper, is an expert who has conducted research related to tilting pad journal bearings during his Ph.D. course in Dr Palazzolo's laboratory and has published several papers in this regard.

Reviewer 3 Report

1.     This study aims to investigate the quantitative performance change due to thermal deformation of the tilting pad journal bearing pad in terms of the change in preload amount. The variable viscosity Reynolds equation and the energy equation were coupled using the relationship between viscosity and temperature, and the solution was obtained using the finite element method. Heat transfer between spinning journal, oil film and pads is considered, and a three-dimensional (3D) finite element (FE) model was used to calculate the thermal deformation of the bearing structure. The preload of tilting pad journal bearings is one of critical design parameters that bearing designers deal with to achieve their goals in the design stage. Overall, the paper structure is complete. The picture and equation are clear and readable.

Author Response

Dear Reviewer,

Thank you very much for your hard work.

Corrections were made according to the opinions of reviewers, and grammatical and expression errors were corrected through a professional English proofreading institution.

Thank you.

Reviewer 4 Report

1. In the theoretical part, the related knowledge of how to consider the thermal deformation of the tilting pad journal bearing pad to calculate bearing stiffness should be added.

2. In Figure 6, the difference increases by more than 0.4 at 1250rpm, but the figure shows only 0.25.

3. How does the viscosity of lubricating oil change with temperature lead to thermal deformation of the tilting pad journal bearing pad?

4. How does the thermal deformation of the tilting pad journal bearing  pad affect the thickness of oil film and fluid pressure?

5. The explanation of curve change trend in Figure 13 (c) is not detailed enough.

Author Response

Dear Reviewer,

Thank you very much for your hard work.

Corrections were made according to the opinions of reviewers, and grammatical and expression errors were corrected through a professional English proofreading institution.

Thank you

1. In the theoretical part, the related knowledge of how to consider the thermal deformation of the tilting pad journal bearing pad to calculate bearing stiffness should be added.

  • Most papers related to numerical analysis of tilting pad journal bearings do not separately deal with the method of calculating bearing dynamic coefficients. This is because the bearing dynamic coefficient calculation method has already been sufficiently covered in existing theoretical books or papers. Therefore, the following sentence was added at the end of the Algorithm section.
  • ‘In this study, the calculation method of the bearing dynamic coefficient evaluated after steady-state prediction is not separately dealt with. This is because the method has already been covered in many existing studies [4,5,17], and most bearing researchers al-ready have sufficient knowledge about it. In this study, a synchronously reduced dynamic coefficient was used.’

2. In Figure 6, the difference increases by more than 0.4 at 1250rpm, but the figure shows only 0.25.

  • It can be seen that the difference between the maximum and minimum values at maximum rpm is 0.4 (maximum value 0.7 - minimum value 0.3 on the scale bar).

3. How does the viscosity of lubricating oil change with temperature lead to thermal deformation of the tilting pad journal bearing pad?

  • As illustrated in Figure 1, viscous shear in the oil film increases the temperature of the lubricating oil, and the heated lubricating oil transfers heat to the bearing pad and journal. Due to this heat transfer, a thermal gradient and thermal deformation are generated in the bearing structure.

4. How does the thermal deformation of the tilting pad journal bearing pad affect the thickness of oil film and fluid pressure?

  • Bearing pads and spinning journal cover a lubricating film area. If thermal deformation occurs in these pads and journal, the film thickness changes. This relationship is described in equation (10), which is the oil film thickness formula.

5. The explanation of curve change trend in Figure 13 (c) is not detailed enough.

  • According to the reviewer's opinion, the explanation about Figure 13 (c) and (d) is changed as following.
  • ‘In the case of the direct damping terms Cyy, (see Fig. 13 (d)), when the bearing thermal deformation is considered, the value is lower than that of the case without the bearing thermal deformation. On the other hand, it can be seen that the stiffness coefficient Cyy in the load direction is not affected much by the thermal deformation of the pad compared to Cxx.’

Reviewer 5 Report

The paper contributes the concept of thermal offset preload and thermal performance preload for investigating the amount of thermal deformation of the bearing pad in terms of bearing performance. Investigating the performance change due to thermal deformation of the tilting pad journal bearing pad in terms of the change in preload amount. There are also some problems, which must be solved before it is considered for publication.

1.English grammar, spelling and sentence structure need to be examined carefully.

2.In Abstract, it is suggested that this article should start with the general background, and then narrow the scope to discuss the related works to be discussed in this article.

3.The relevant research in the introduction is earlier and needs to be supplemented by the relevant research in recent years. You should quote all the files you use correctly. The recommended references are as follows:

[1] Guangwu Zhou, Jinsheng Qiao, Wei Pu, Ping Zhong. Analysis of mixed lubrication performance of water-lubricated rubber tilting pad journal bearing[J]. Tribology International, 2022, 169: 107423.

[2] Jianlin Cai, Yanfeng Han, Guo Xiang, Jiaxu Wang, and Liwu Wang. Effects of wear and shaft-shape error defects on the tribo-dynamic response of water-lubricated bearings under propeller disturbance[J]. Physics of Fluids, 2022, 34(7): 077118.

[3] Jianlin Cai, Yanfeng Han, Guo Xaing, Cheng Wang, Liwu, Wang, Shouan, Chen. Influence of the mass conservation cavitation boundary on the tribodynamic responses of the micro-groove water-lubricated bearing[J]. Surface Topography: Metrology and Properties, 2022, (10): 045011.

4.The figures in your paper are a little vague. Consider replacing them with clearer one.

5.Another obvious problem in this paper is the lack of sufficient explanation for simulation. For example, the bearing selected in line 378, why is it selected? Is it universal? At the same time, you need to explain in detail your simulation results and why you get such results.

6.Please determine the format of the tables and make sure they are consistent.

7.The description of lines 461-465 can not support the correctness of this model. It is suggested that adopt the published experimental results of and theoretical derivation of tilting pad journal bearing

8.The Conclusion is lack of theoretical and experimental support. The advantages of offset thermal preload and performance thermal preload are not emphasized enough in conclusion (b), there is no data support for the ratio in conclusion (d), and the damping coefficient is not mentioned in conclusion (e).

9.It is suggested that the formula be put together with the corresponding comments in the text.

10.The innovation point is not clear enough. The significance of this paper is not expounded sufficiently. The author needs to highlight this paper's innovative contributions.

Author Response

Dear Reviewer,

Thank you very much for your hard work.

Corrections were made according to the opinions of reviewers, and grammatical and expression errors were corrected through a professional English proofreading institution.

Thank you

1.English grammar, spelling and sentence structure need to be examined carefully.

  • According to the reviewer's opinion, this paper was edited by a professional English proofreading institution. Thank you.

2.In Abstract, it is suggested that this article should start with the general background, and then narrow the scope to discuss the related works to be discussed in this article.

  • Some researchers start the abstract with a general background, while some researchers list the research results from the beginning of the abstract. Due to the limitation of the number of abstract words, the authors listed only the contents of the research, and the explanation of the background knowledge was explained in the introduction. Thank you.

3.The relevant research in the introduction is earlier and needs to be supplemented by the relevant research in recent years. You should quote all the files you use correctly. The recommended references are as follows:

[1] Guangwu Zhou, Jinsheng Qiao, Wei Pu, Ping Zhong. Analysis of mixed lubrication performance of water-lubricated rubber tilting pad journal bearing[J]. Tribology International, 2022, 169: 107423.

[2] Jianlin Cai, Yanfeng Han, Guo Xiang, Jiaxu Wang, and Liwu Wang. Effects of wear and shaft-shape error defects on the tribo-dynamic response of water-lubricated bearings under propeller disturbance[J]. Physics of Fluids, 2022, 34(7): 077118.

[3] Jianlin Cai, Yanfeng Han, Guo Xaing, Cheng Wang, Liwu, Wang, Shouan, Chen. Influence of the mass conservation cavitation boundary on the tribodynamic responses of the micro-groove water-lubricated bearing[J]. Surface Topography: Metrology and Properties, 2022, (10): 045011.

  • This study is related to the viscous shear that occurs in the oil film and the resulting thermal deformation of the structure. The papers suggested as references are not included as references because they did not consider thermal effects. However, since these papers provide excellent research results, the authors will use them as references for future research. Thanks for the advice.

4.The figures in your paper are a little vague. Consider replacing them with clearer one.

  • In accordance with reviewer’s suggestion, Figure size is increased

5.Another obvious problem in this paper is the lack of sufficient explanation for simulation. For example, the bearing selected in line 378, why is it selected? Is it universal? At the same time, you need to explain in detail your simulation results and why you get such results.

  • The bearing model used in this study is a model in which the size of a bearing model used in a power plant hot water pump is changed with the same Sommerfeld number. It is also a model used in the previous papers of the corresponding author of this paper. Details of where the bearing model is used were not mentioned at the request of the bearing manufacturer. The following sentence is added in the paper.
  • ‘Details of where the bearing model is used were not mentioned at the request of the bearing manufacturer’

6.Please determine the format of the tables and make sure they are consistent.

  • The format of the table has been modified according to the advice given by the reviewers. Thank you.  

7.The description of lines 461-465 can not support the correctness of this model. It is suggested that adopt the published experimental results of and theoretical derivation of tilting pad journal bearing

  • Pad deformation pad can be measured through experiments. However, since the deformation of the pad reflects both thermal and elastic deformation, it is impossible to measure the thermal deformation of the pad simultaneously experimentally.
  • Also, in previous studies, there have been studies that predicted the thermal deformation of the pad through finite element models; however, there is no paper presenting the thermal deformation data distributed on the pad like this paper.
  • Therefore, there is no method to verify the results of this thermal deformation analysis through other references. Following sentence is added.
  •  ‘The numerical model used in this study has already been verified in previous study by the corresponding author of this paper [41].’

8.The Conclusion is lack of theoretical and experimental support. The advantages of offset thermal preload and performance thermal preload are not emphasized enough in conclusion (b), there is no data support for the ratio in conclusion (d), and the damping coefficient is not mentioned in conclusion (e).

  • Following sentences are added to emphasize the thermal preload
  • (a)          The concept of the offset thermal preload was suggested to quantify the thermal deformation shape of the bearing pad.
  • (b)          The concept of the performance thermal preload was suggested to quantify the thermal deformation of the bearing structure and the resulting bearing performance change.
  • Following sentences are added to show data support the ratio and mention damping coefficients.
  • (c)          The model neglecting the pad thermal deformation showed a stiffness coefficient up 92% higher than that of the model not considering the thermal deformation.
  • (d)          When the pad thermal deformation was considered, the damping coefficient was up to 10% higher than when it was not taken into account; however, the effect seems insignificant compared to the increase in the stiffness coefficient.

9.It is suggested that the formula be put together with the corresponding comments in the text.

  • Formulas are rearranged according to the reviewer’s suggestion.

10.The innovation point is not clear enough. The significance of this paper is not expounded sufficiently. The author needs to highlight this paper's innovative contributions.

  • Following sentences are added in the ‘4. Conclusions’
  • The innovation points of this study compared to previous studies are as follows.
  • (a)          The concepts of the offset thermal preload and the performance thermal preload were suggested to quantify the thermal deformation of the bearing structure and the resulting bearing performance change.
  • (b)          It was found that as the temperature of the bearing increases, the performance change of the bearing due to the thermal deformation of the pad also increases.

Round 2

Reviewer 2 Report

Authors have dismissed the reviewer's main comment. Knowing that one of the co-authors has worked with Dr. Palazzolo in the same topic makes more glaring the deficiency. Since the early 1990s', the analyses and codes produced by Dr. Palazzolo and co-workers (former students) have included both thermal and mechanical deformations of the pads (and pivots) on TPJB performance. Please note there is NO sound justification to just include thermal deformations and to ignore pressure deformations. The claim that the mechanical deformations are forthcoming is just lazy. The manuscript, in spite of its length and pretty pictures, offers little to new knowledge. In addition, the absence of any verification against experimental results makes the work suspicious. Please add a modicum of comparisons, in particular for force coefficients. There is abundant experimental data from Childs and San Andres at Texas A&M, Hagemann at Clausthal, Penacchio in Italy. Golden oldies from Dmochowski, Fillon, Nicholas, etc could also serve for validation.  

Author Response

Dear Reviewer, 
All authors of this paper are very grateful for the notes provided by the reviewer. 
We have made the following corrections to your comments, and we are deeply grateful for the significant improvement in the quality of this paper.

1. It is true that the results of the research conducted by Palazzolo and his students included thermal and sometimes elastic deformation of the pad.
However, the research results at the time did not mention the details of the thermal deformation shape of the bearing pad, and there was no attempt to quantify this thermal deformation shape and relate it to the change in bearing performance.
The contribution of this study is that it tries to correlate the thermal deformation shape of bearing pads, which has not been attempted before, with a quantitative indicator called thermal preload change, and then correlates it with bearing performance change.
To emphasize this point, the following was added to the beginning of Abstract and the end of Introduction.

--> beginning of Abstract
'Thermal deformation of journal bearings operating under high-temperature conditions can have a significant effect on changes in bearing performance. However, no attempt has been made to quantify this amount of thermal deformation and link it to the performance change of the bearing. '

--> end of Introduction
' Many studies have been conducted considering the thermal effect that occurs in journal bearings [3-43]. What these previous studies have in common is that the viscous shear heat generated in the oil film changes the lubricant viscosity, and the changed viscosity has a significant effects on the bearing performance. In addition, a few more advanced studies [3, 4, 37, 38] predicted the thermal deformation of the bearing structure and the resulting change in oil film thickness using 3D finite element model. However, there has been no study to correlate the specific thermal deformation shape of the bearing structure with the performance change of the bearing by quantifying it. Although Suh and Palazzolo [4] proposed the concept of thermal preload, it was not studied in terms of bearing performance change.'

2. This study focuses on the performance change due to thermal deformation of the bearing and does not deal with the performance change due to elasticity change. This is because thermal and elastic deformation occur at the same time, but their causes are completely different, and their shapes are also very different.
In addition, the follow-up study of this study by the corresponding author is a study on the performance change due to the elastic deformation of the bearing pad. Accordingly, the following content has been added to the 400th line of this paper. 

--> 'According to the experience of the corresponding author of this study, it is expected that the unit load of 1.41 MPa can cause some performance change due to the pad elastic de-formation. This performance change can vary greatly depending on the shape and thick-ness of the pad. Since this study deals with the performance change due to thermal de-formation of the bearing pad, study on the performance change due to the pad elastic deformation is considered to be outside the scope of this study. Also, as the follow-up study to this study is the performance change due to the pad elastic deformation, it will be dealt with in detail in the next study.'

3. We agree that the code used in this thesis needs to be verified, but since this code has already been verified in a study by the corresponding author two years ago, it is thought that additional verification is unnecessary, and the following is mentioned in line 420 of this thesis.

--> 'The numerical model used in this study has already been verified in previous study per-formed by the corresponding author of this paper [41].'